# IL-6 receptor blockade corrects defects of XIAP-deficient regulatory T cells

Wan-Chen Hsieh[1], Tzu-Sheng Hsu[1], Ya-Jen Chang[2] & Ming-Zong Lai [1]

X-linked lymphoproliferative syndrome type-2 (XLP-2) is a primary immunodeficiency disease attributed to XIAP mutation and is triggered by infection. Here, we show that mouse $Xiap^{-/-}$ regulatory T (Treg) cells and human XIAP-deficient Treg cells are defective in suppressive function. The $Xiap^{-/-}$ Treg cell defect is linked partly to decreased SOCS1 expression. XIAP binds SOCS1 and promotes SOCS1 stabilization. Foxp3 stability is reduced in $Xiap^{-/-}$ Treg cells. In addition, $Xiap^{-/-}$ Treg cells are prone to IFN-γ secretion. Transfer of wild-type Treg cells partly rescues infection-induced inflammation in $Xiap^{-/-}$ mice. Notably, inflammation-induced reprogramming of $Xiap^{-/-}$ Treg cells can be prevented by blockade of the IL-6 receptor (IL-6R), and a combination of anti-IL-6R and $Xiap^{-/-}$ Treg cells confers survival to inflammatory infection in $Xiap^{-/-}$ mice. Our results suggest that XLP-2 can be corrected by combination treatment with autologous iTreg (induced Treg) cells and anti-IL-6R antibody, bypassing the necessity to transduce Treg cells with XIAP.

---

[1] Institute of Molecular Biology, Academia Sinica, Academia Sinica, Taipei 11529, Taiwan. [2] Institute of Biomedical Sciences, Academia Sinica, Taipei 11529, Taiwan. Correspondence and requests for materials should be addressed to M.-Z.L. (email: mblai@gate.sinica.edu.tw)

X-linked lymphoproliferative syndrome type-2 (XLP-2) is a primary immunodeficiency disease linked to the absence of X-linked inhibitor of apoptosis protein (XIAP) due to *XIAP* gene mutation[1–3]. XIAP is a member of the inhibitor of apoptosis protein (IAP) family that directly inhibits caspase-3, caspase-7 and caspase-9, and transmits caspase-independent signals[4, 5]. Like many other primary immunodeficiency diseases that are associated with recurrent and serious infection, XLP-2 is mostly driven by Epstein-Barr virus (EBV). Infection-induced inflammation can cause life-threatening lymphoproliferative disorder in XLP-2, manifested by excess cytokine production from over-activated lymphocytes and macrophages (known as hemophagocytic lymphohistiocytosis), with recurrent splenomegaly, fever, and hemorrhagic colitis[6, 7]. XIAP mutation is also found in a small subpopulation of patients with pediatric-onset Crohn's disease[8]. In contrast to patients with XLP-2, $Xiap^{-/-}$ mice housed in specific pathogen free (SPF) environments seem normal and do not exhibit a phenotype reminiscent of XLP-2[9–11]. However, upon infection with *Listeria monocytogenes*, *Chlamydophila pneumoniae*, *Candida albicans* or murine γ-herpesvirus 68, $Xiap^{-/-}$ mice develop hyper-inflammation and an XLP-2-like syndrome, with resulting high mortality[12–15]. Similar to patients with XLP-2, $Xiap^{-/-}$ mice are vulnerable to infection-induced inflammation. The development and activation of T cells is mostly normal in $Xiap^{-/-}$ mice[8, 9, 11, 15]. Impaired innate immune responses are found in $Xiap^{-/-}$ mice infected with these selective pathogens[12, 13, 15]. XIAP-deficiency does not affect TNF-, LPS- or Pam3CSK4-induced NF-κB activation, but results in selective defects of BCL10- and NOD-mediated NF-κB activation in myeloid cells[8, 14–19]. Similar to other primary immunodeficiency diseases with specific mutations in pattern recognition receptors[20–22], XLP-2 is probably an innate immunodeficiency disease caused by defective signaling downstream of the dectin-1 and NOD receptors[2]. Despite our knowledge being advanced by these studies in $Xiap^{-/-}$ myeloid cells, the molecular mechanism of how mutation in XIAP leads to excess lymphocyte activation in XLP-2 patients is incompletely understood[6].

Regulatory T (Treg) cells suppress excess T cell activation and maintain peripheral T cell tolerance. Treg cells are classified into thymus-derived CD4+ CD25+Foxp3+ regulatory T (tTreg) cells, induced regulatory T cells (iTreg cells), and peripherally generated Treg cells (pTreg cells)[23]. The development and function of Treg cells is controlled by Foxp3 expression[24]. Foxp3 is regulated at epigenetic, transcriptional, and posttranslational levels[25]. Loss of Foxp3 during the lifetime of Treg cells ablates the suppressive activity[26, 27]. Within inflammatory environments and accompanied by loss of Foxp3 expression, Treg cells can be converted into effector T cells such as T helper type 1 (Th1)[28–30], Th17[31, 32], follicular B helper T (T$_{FH}$)[33], or Th2 cells[34]. These reprogrammed Treg cells contribute to pathology and are targets of immunotherapy[35]. Treg cell plasticity is regulated by a long list of molecules, including suppressor of cytokine signaling 1 (SOCS1), which is one of the transcription factors required to maintain the inhibitory function of these cells[36]. SOCS1 deficiency leads to over-activation of signal transducer and activator transcription 1 (STAT1) and STAT3 in Treg cells[36–38], resulting in excess inflammatory signaling, loss of Foxp3 expression, and spontaneous autoimmunity[36].

Adoptive transfer of Treg cells has been explored as a potential therapy for various autoimmune diseases, graft vs host disease, as well as transplantation rejection[39–41]. Autologous Treg cell numbers can be expanded ex vivo to a high number and their safe application in vivo has been confirmed[42]. A further development is the genetic modification of Treg cells for expression of a chimeric antigen receptor (CAR)[43] or antigen-specific T cell receptor (TCR)[41, 44]. Similarly, defective Treg cells caused by mutations in specific genes can be corrected by re-introduction of the respective wild-type gene.

In the present study, we demonstrate that XIAP is required for the suppressive function of Treg cells. XIAP-deficient Treg cells are ineffective in inhibiting inflammation. SOCS1 expression is reduced in XIAP-deficient Treg cells. XIAP promotes SOCS1 K63 ubiquitination and maintains SOCS1 protein stability. Transfer of wild-type Treg cells partly suppresses infection-induced inflammation in $Xiap^{-/-}$ mice. Moreover, a combination of $Xiap^{-/-}$ Treg cells and anti-IL-6R corrects the vulnerability of $Xiap^{-/-}$ mice to infection. Our results provide evidence of a mechanism underlying the generation of XLP-2 syndromes in XIAP mutant patients. Furthermore, we demonstrate the therapeutic feasibility of combining autologous Treg cells and anti-IL-6R for the treatment of primary immunodeficiency diseases such as XLP-2.

## Results

**Impaired inhibitory activity of $Xiap^{-/-}$ Treg cells.** A previous study showed that XIAP-deficiency did not affect the development of tTreg cells and that the frequency of CD4+Foxp3+ cells was comparable between $Xiap^{-/-}$ mice and littermate control (WT) animals (Fig. 1a)[15]. Generation of iTreg cells was similar between WT and $Xiap^{-/-}$ naive CD4+CD25− T cells when time and dosage of TGF-β were optimized (Fig. 1a). We also determined the expression of several Treg cells-associated molecules in isolated WT and $Xiap^{-/-}$ tTreg cells (Supplementary Figure 1) and found that expressions of CTLA-4, GITR, LAG3, and FR4 were comparable between WT and $Xiap^{-/-}$ tTreg cells (Fig. 1b). Production of IL-10 and TGF-β in $Xiap^{-/-}$ tTreg cells was indistinguishable from that in WT tTreg cells (Fig. 1c). Despite having normal phenotypes, $Xiap^{-/-}$ tTreg cells were less effective than WT tTreg cells in suppressing the proliferation of CD4+CD25− effector T cells (Fig. 1d). Inhibition of CD4+CD25− T cell activation by $Xiap^{-/-}$ iTreg cells was also compromised (Supplementary Figure 2). Since XIAP is an anti-apoptotic and signaling protein, the proliferation and viability of $Xiap^{-/-}$ tTreg cells were examined and found to be comparable with WT tTreg cells (Supplementary Figure 3). We also used human XIAP-knockdown Treg cells to mimic Treg cells of XLP-2 patients (Fig. 1e). The differentiation of human XIAP-deficient iTreg cells was similar to that of control iTreg cells (Supplementary Figure 4). However, the suppressive activity of human XIAP-knockdown tTreg cells was impaired relative to control tTreg cells (Fig. 1f, Supplementary Figure 5).

$Xiap^{-/-}$ tTreg cells also displayed a diminished capacity to suppress T cell activation in an in vivo functional assay. Colitis was induced in $Rag1^{-/-}$ mice by administration of CD4+CD25− T cells, leading to body weight loss (Fig. 1g), rectal prolapse and diarrhea. Tissue sections of inflamed colon revealed inflammatory infiltrate, crypt cell damage and goblet cell damage (Fig. 1h, Teff). Co-administration of WT tTreg cells effectively suppressed the induction of colitis by CD4+CD25− T cells. In contrast, co-administration of $Xiap^{-/-}$ tTreg cells did not prevent the colitis-associated pathogenesis (Fig. 1g and h), suggesting that the absence of XIAP greatly diminished the suppressive activities of tTreg cells in vivo. Therefore, even though XIAP is not involved in the development of tTreg cells or iTreg cell differentiation, XIAP is essential for the suppressive function of Treg cells in vitro and in vivo.

**XIAP-deficiency decreases Foxp3 stability in Treg cells.** We next examined why the inhibitory activities of $Xiap^{-/-}$ Treg cells were defective. $Xiap^{-/-}$ tTreg cells were stimulated through TCR/CD28 in the presence of IL-2, and the expression of Foxp3 was determined. Total numbers of Foxp3+ T cells were reduced in

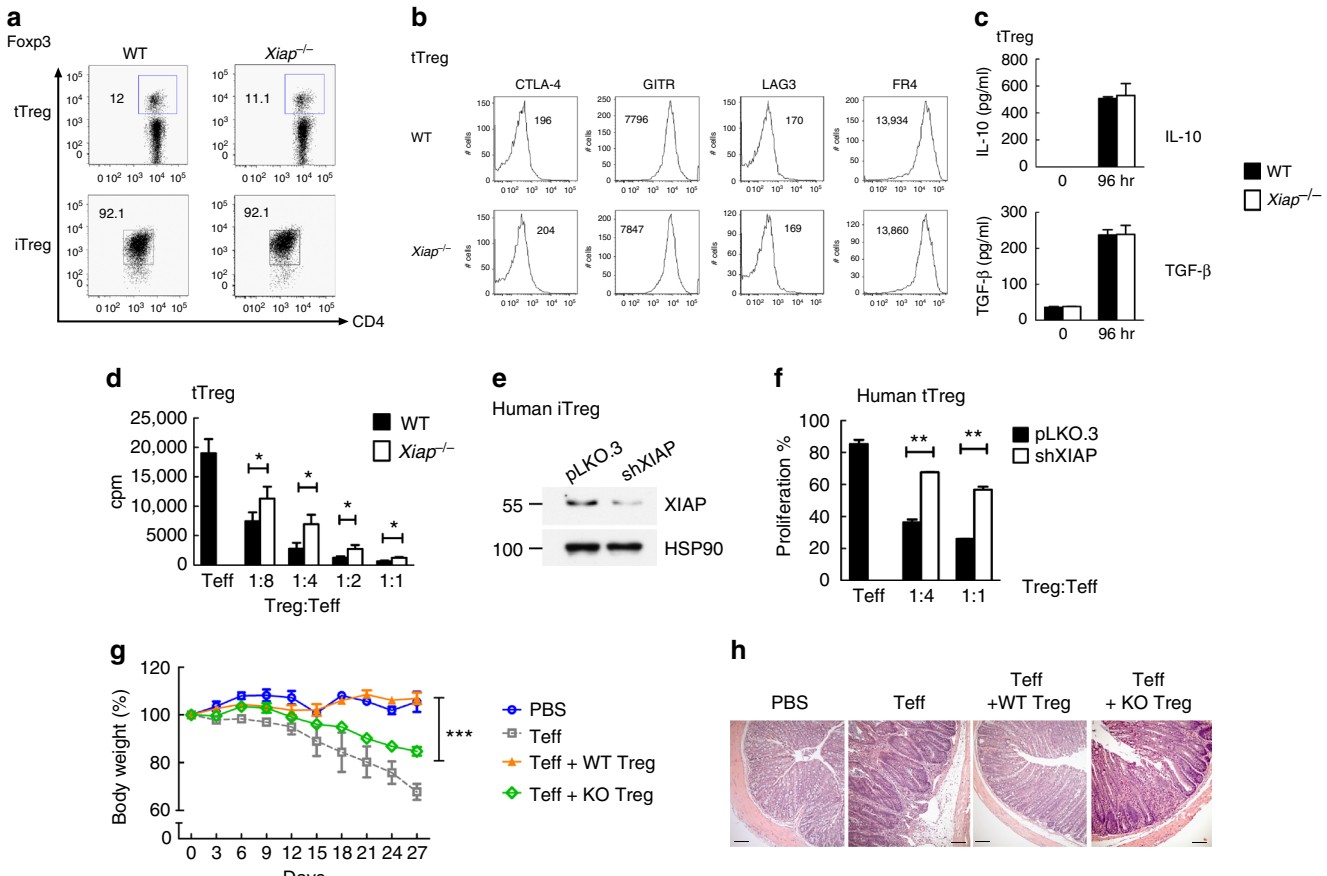

**Fig. 1** XIAP-deficiency impairs Treg cell suppressive function. **a** Comparable control and *Xiap*⁻/⁻ tTreg cells and iTreg cells development. The fractions of splenic CD4⁺Foxp3⁺ population from control (WT) and *Xiap*⁻/⁻ mice were determined (tTreg). WT and *Xiap*⁻/⁻ CD4⁺CD25⁻ T cells were treated with anti-CD3/CD28 plus TGF-β and IL-2, and Foxp3 expression at day 5 was assessed (iTreg). Experiments were independently repeated six times. **b** XIAP-deficiency does not affect the Treg cells phenotype. Expressions of CTLA-4, GITR, LAG3 and FR4 in WT and *Xiap*⁻/⁻ tTreg cells were determined. Numbers indicate mean fluorescence intensities. **c** Normal IL-10 and TGF-β production in *Xiap*⁻/⁻ tTreg cells. CD4⁺CD25⁺ cells were stimulated with anti-CD3/CD28 and IL-2 for 96 h, before generation of IL-10 and TGF-β was determined. **d** Impaired in vitro suppressive activity of *Xiap*⁻/⁻ tTreg cells. CD4⁺CD25⁻ cells were incubated with anti-CD3, antigen-presenting cells, and the indicated ratios of WT and *Xiap*⁻/⁻ CD4⁺CD25⁺ tTreg cells. [³H]thymidine incorporation was determined at 80 h. Values are mean ± SD of triplicate samples in an experiment. *$P < 0.05$, **$P < 0.01$ for unpaired *t*-test. Experiments were reproduced independently three times with similar outcomes. **e** Knockdown of XIAP in human Treg cells. The levels of XIAP in control (pLKO.3) and XIAP-knockdown human iTreg cells were determined by immunoblots. **f** Impaired suppressive activity in XIAP-deficient human tTreg cells. Human effector T cells (Teff) were labeled with 2 μM CFSE, activated by anti-CD3/CD28 in the presence of the indicated ratio of human control and XIAP-knockdown tTreg cells, and collected at 72 h. Proliferation was determined by gating CD4⁺ T cells for their CFSE intensity in flow cytometry. **g, h** Diminished inhibitory activity of *Xiap*⁻/⁻ tTreg cells in vivo. CD4⁺CD25⁻ effector T cells were administered intraperitoneally into *Rag1*⁻/⁻ mice with WT or *Xiap*⁻/⁻ tTreg cells. Body weight was determined at the indicated time-points (**g**). Data are the mean ± SD of six mice in each group. ***$P < 0.001$ for two-way ANOVA. Mice were killed at day 27 and colons were removed, fixed in paraformaldehyde, sectioned, and stained with H&E. Micrographs are representative of the six mice in each group. Bar indicates 100 μm

*Xiap*⁻/⁻ tTreg cells (Fig. 2a and b). We also determined Foxp3 levels in the adoptively transferred CD45.1⁺ tTreg cells isolated from *Rag1*⁻/⁻ mice (Fig. 1f). Figure 2c illustrates that the Foxp3^high fraction in *Xiap*⁻/⁻ tTreg cells activated in vivo were lower than those in WT tTreg cells. Quantitation of the Foxp3⁺ population confirmed a nearly two-fold reduction in the Foxp3^high fraction in *Xiap*⁻/⁻ tTreg cells (Fig. 2d). Thus, XIAP-deficiency decreases Foxp3 stability in Treg cells in vitro and in vivo.

**XIAP interacts with SOCS1 and increases SOCS1 expression.** A recent report indicated that SOCS1 is essential for Foxp3 stability and its suppressive function[36]. We examined IL-2-stimulated SOCS1 expression in WT and XIAP-deficient T cells. SOCS1 was decreased in IL-2-treated *Xiap*⁻/⁻ T-cells (Fig. 3a). XIAP-

knockdown also decreased SOCS1 levels in human iTreg cells (Fig. 3b). IL-2-induced *Socs1* transcript levels in T cells were not affected by XIAP-deficiency (Supplementary Figure 6a). In contrast, the protein stability of SOCS1 was dependent on the presence of XIAP (Supplementary Figure 6b), which is supported by the enhanced SOCS1 expression under increased levels of XIAP (Fig. 3c) and that XIAP knockdown in human iTreg cells increased cycloheximide-induced SOCS1 degradation (Supplementary Figure 6c). These results suggest that XIAP regulates SOCS1 expression by maintaining SOCS1 protein stability.

We found an association between SOCS1 and XIAP. Immunoprecipitation of SOCS1-HA brought down XIAP-FLAG (Fig. 3d), and precipitation of endogenous SOCS1 pulled down endogenous XIAP in T cells (Fig. 3e). XIAP consists of N-terminal baculovirus IAP (BIR) 1, BIR2 and BIR3, as well as

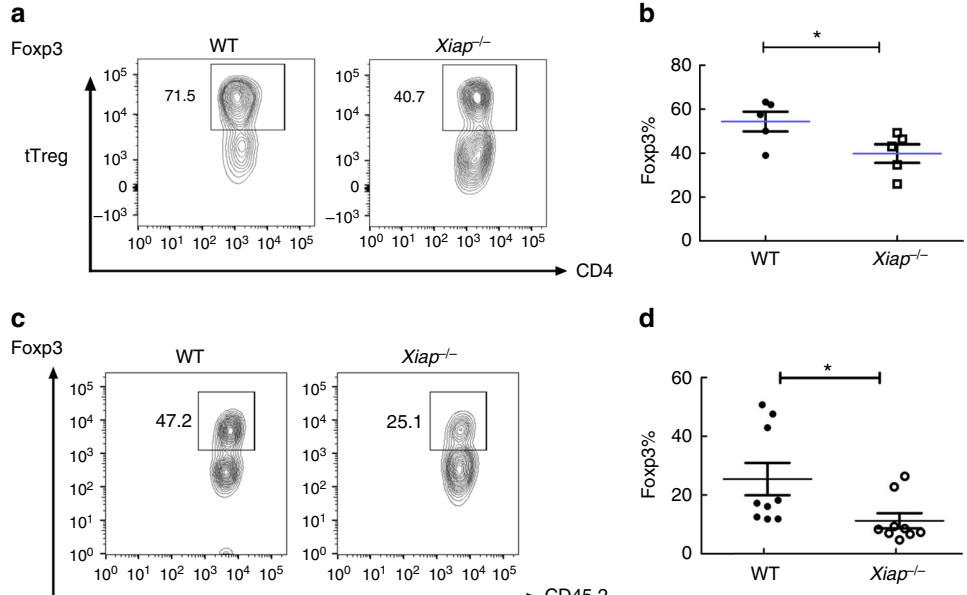

**Fig. 2** Foxp3 instability in activated $Xiap^{-/-}$ Treg cells. **a, b** Reduced Foxp3$^+$ population in activated $Xiap^{-/-}$ tTreg cells. WT and $Xiap^{-/-}$ tTreg cells were stimulated with anti-CD3/CD28 in the presence of IL-2 for 4 days. Foxp3 expression of activated tTreg cells was determined by intracellular staining and analyzed by flow cytometry. The gated section represents the Foxp3$^+$ population and the number indicates the percentage of each population (**a**). The Foxp3$^+$ fractions in WT and $Xiap^{-/-}$ tTreg cells were quantitated (**b**), $n = 5$. **c, d** Diminished Foxp3$^{high}$ population in adoptively transferred $Xiap^{-/-}$ tTreg cells. CD45.2$^+$ tTreg cells were recovered from CD45.1$^+$ $Rag1^{-/-}$ mice into which CD45.2$^+$ tTreg cells and CD45.1$^+$ CD4$^+$CD25$^-$ T cells had been transferred a month earlier. The isolated CD45.2$^+$ CD4$^+$ T cells were reactivated with TPA/A23187 and Foxp3 expression was determined by intracellular staining (**c**). The Foxp3$^+$ fractions in the transferred WT and $Xiap^{-/-}$ tTreg cells re-isolated from $Rag1^{-/-}$ mice were quantitated (**d**), $n = 9$. *$P < 0.05$ for unpaired t-test

a C-terminal really interesting new gene (RING)-finger domain. Using different truncated forms of FLAG-tagged XIAP, we mapped the BIR1 domain of XIAP as being the SOCS1-interacting region (Fig. 3f). For SOCS1, which comprises an N-terminus, a central Src homology 2 (SH2) domain and a C-terminal SOCS-BOX domain, we found the SH2 domain to be the XIAP-binding region (Fig. 3g).

**XIAP promotes SOCS1 K63 ubiquitination**. Previous reports have found that SOCS1 is associated with the Elongin B/C complex, which functions as an E3 ligase. Immunoprecipitation of overexpressed Elongin B/C brought down SOCS1-HA (Fig. 4a). Notably, co-expression of full-length XIAP-FLAG increased the association of SOCS1-HA with the Elongin B/C-Myc complex (Fig. 4a). By contrast, ΔRING-XIAP did not enhance association of SOCS1 with Elongin B/C (Fig. 4a). We also determined whether XIAP promoted SOCS1 poly-ubiquitination. Co-expression of XIAP enhanced the addition of WT ubiquitin or K63 ubiquitin, but not K48 ubiquitin, to SOCS1 (Fig. 4b). In an in vitro ubiquitination analysis, addition of recombinant XIAP (but not XIAPΔRF) to reaction mixtures containing ubiquitin, E1, E2 (UBC13), Elongin B/C and recombinant SOCS1 increased K63 ubiquitination of SOCS1 (Fig. 4c). Together, these results suggest that XIAP binds SOCS1 and promotes SOCS1 K63 polyubiquitination, likely contributing to the increased protein stability of SOCS1.

***Xiap*$^{-/-}$ Treg cells are prone to IFN-γ and IL-17 production**. We examined whether reduced SOCS1 in $Xiap^{-/-}$ Treg cells conferred on them the susceptibility to produce inflammatory cytokines, similar to $Socs1^{-/-}$ Treg cells. Foxp3-GFP tagged WT and $Xiap^{-/-}$ tTreg cells, isolated by GFP expression, were

activated by CD3/CD28 plus IL-2, with the additional presence of IL-12 or IL-1α/IL-1β/IL-6. We found a smaller increase in IFN-γ expression for $Xiap^{-/-}$ tTreg cells after TCR re-stimulation (Fig. 5a). The presence of IL-12 substantially increased the fraction of IFN-γ-expressing $Xiap^{-/-}$ tTreg cells relative to WT tTreg cells (Fig. 5a and b), and elicited a significant increase in IFN-γ secretion by reactivated $Xiap^{-/-}$ tTreg cells (Fig. 5c). Similarly, human XIAP-knockdown iTreg cells generated more IFN-γ after IL-12 co-stimulation than WT iTreg cells (Fig. 5d and e). Co-treatment with IL-6 also led to enhanced production of IFN-γ in human XIAP-deficient iTreg cells (Fig. 5d).

Enhanced generation of IL-17 was also detected in $Xiap^{-/-}$ tTreg cells compared to WT tTreg cells after being co-stimulated with TCR and IL-6/IL-1 (Fig. 5f). We further examined the cytokine profile of tTreg cells isolated from $Rag1^{-/-}$ mice after induction of colitis by effector T cells. The recovered CD45.2$^+$ T cells, representing the transferred tTreg cells, were assessed for expression of IFN-γ and IL-17A (Fig. 5g and h). Increased IFN-γ and IL-17A expression was observed in $Xiap^{-/-}$ tTreg cells isolated from $Rag1^{-/-}$ mice, illustrating the enhanced plasticity of $Xiap^{-/-}$ tTreg cells in vivo.

**Increased STAT1 and STAT3 activation in $Xiap^{-/-}$ Treg cells**. Inflammatory cytokines including IL-12, IL-1, and IL-6 trigger JAK-STAT signaling in T cells. SOCS1 is a negative regulator of JAK-STAT signaling, so we determined whether the reduced SOCS1 in $Xiap^{-/-}$ iTreg cells led to an enhanced response to inflammatory cytokines. WT and $Xiap^{-/-}$ iTreg cells were stimulated with IL-12 or IL-1 plus IL-6, and we found increased phosphorylation of STAT1 or STAT3 in $Xiap^{-/-}$ iTreg cells relative to WT iTreg cells, in the context of comparable levels of STAT1 and STAT3 (Supplementary Figure 7a and b). Therefore, XIAP deficiency leads to enhanced activation of Treg cells in

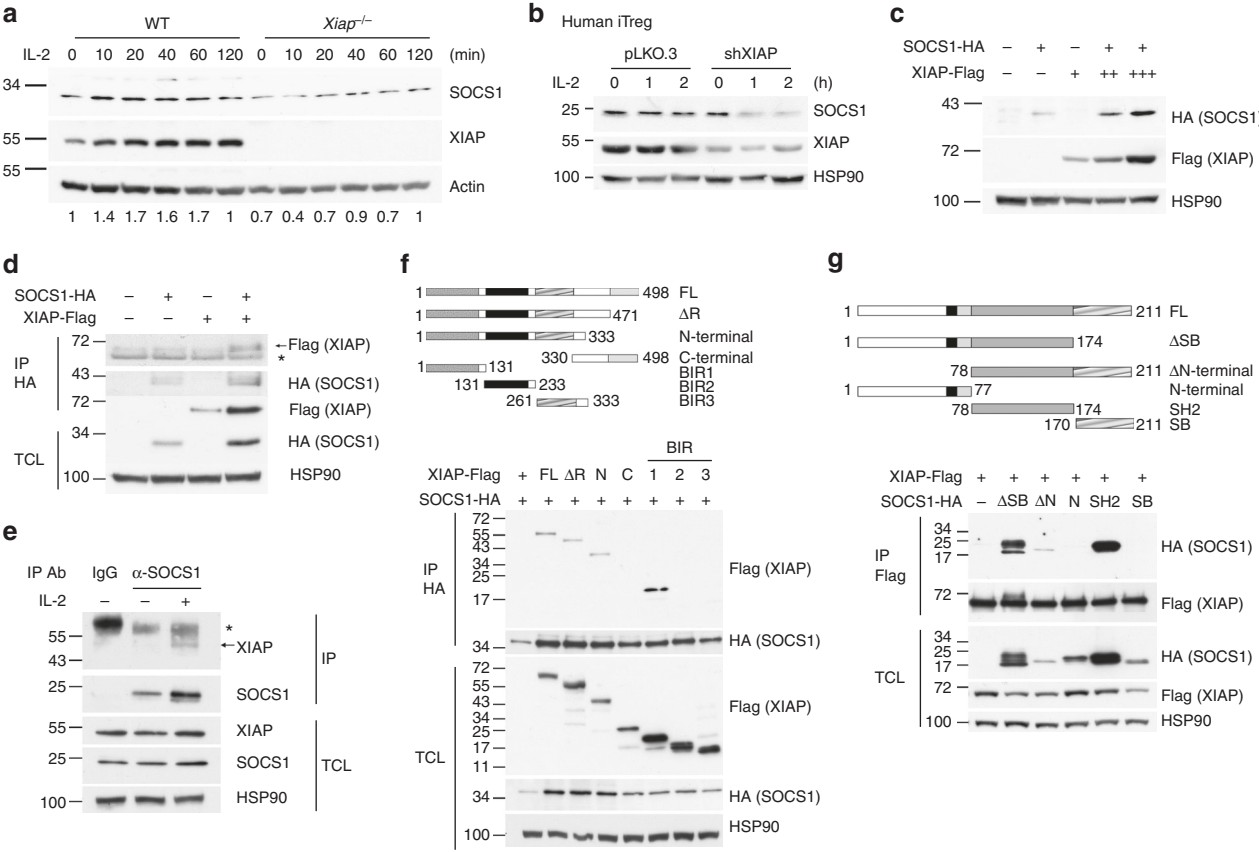

**Fig. 3** XIAP interacts with SOCS1 and enhances SOCS1 expression. **a** XIAP deficiency impairs IL-2-induced SOCS1 expression. T cells were collected at 0, 10, 20, 40, 60, and 120 min after IL-2 treatment. SOCS1 expression of lysate was detected by anti-SOCS1. Protein levels were quantitated by densitometry and normalized by actin control. The level of SOCS1 in WT T cells was used as 1 for comparison. **b** XIAP-deficiency decreases SOCS1 expression in human iTreg cells. Control and XIAP-knockdown human iTreg cells were treated with IL-2 and the levels of SOCS1 were determined at the indicated time-points. **c** XIAP enhances SOCS1 expression. XIAP-FLAG and SOCS1-HA were co-transfected into HEK293T cells. After 24 h of transfection, cell lysates were prepared and SOCS1 and XIAP expression was determined with anti-FLAG and anti-HA. **d** XIAP interacts with SOCS1. XIAP-FLAG and SOCS1-HA were co-transfected into HEK293T cells as indicated. Total cell lysates were immunoprecipitated by anti-HA and the presence of SOCS1 and XIAP-FLAG in the precipitates and lysates was determined. * indicates immunoglobulin heavy chain. **e** Endogenous XIAP interacts with SOCS1. Mouse peripheral T cells from spleen and lymph nodes were treated with IL-2 as indicated and then 600 μg of cell lysates were immunoprecipitated with anti-SOCS1 or control goat IgG. The contents of endogenous XIAP were determined. * indicates immunoglobulin heavy chain. **f** The BIR1 domain of XIAP interacts with SOCS1. Full-length (FL), RING domain-deleted (ΔR), N-terminus (N), C-terminus (C), BIR1, BIR2 or BIR3 of XIAP-FLAG were co-transfected with SOCS1-HA into HEK293T cells. Total cell lysates were immunoprecipitated with anti-HA and the presence of XIAP variants and SOCS1 in the pull-down complex and cell lysates was determined. **g** The SH2 domain of SOCS1 binds XIAP. Full-length (FL), SOSC box-deleted (ΔSB), N-terminal-deleted (ΔN), N-terminal (N), SH2 domain, or SOCS box (SB) of SOCS1 were transfected with XIAP-FLAG into HEK293T cells as indicated. Total cell lysates were immunoprecipitated by anti-FLAG and the presence of SOCS1 variants and XIAP in the precipitates and cell lysates was determined. Each experiment (**a**, **c–g**) was independently repeated three times with similar results

response to inflammatory cytokines, similar to *Socs1*[−/−] Treg cells[36, 37,]. Since SOCS1 inhibits the activation of IL-2-STAT5[45], IL-2-induced phosphorylation of STAT5 was also increased in *Xiap*[−/−] iTreg cells (Supplementary Figure 8).

**Introduction of SOCS1 corrects defects in *Xiap*[−/−] Treg cells**. We next examined whether expression of SOCS1 restored the function of *Xiap*[−/−] Treg cells. HA-SOCS1 was re-introduced into *Xiap*[−/−] Treg cells by retroviral transduction (Supplementary Figure 9a). Expression of HA-SOCS1 restored the impaired in vitro suppressive activity of *Xiap*[−/−] iTreg cells (Supplementary Figure 9b). The enhanced production of IFN-γ stimulated by IL-12 or IL-6/IL-1, as well as the increased generation of IL-17 activated by IL-1/IL-6, in *Xiap*[−/−] iTreg cells was suppressed by the re-introduction of HA-SOCS1 (Supplementary Figure 9c and d). Expression of SOCS1 also increased FOXP3 in *Xiap*[−/−] tTreg

cells to levels comparable to WT tTreg cells (Supplementary Figure 9e), indicating enhanced Foxp3 stability. IL-12-mediated IFN-γ production in *Xiap*[−/−] tTreg cells was similarly inhibited by SOCS1 re-introduction (Supplementary Figure 9f). Altogether, these results suggest that SOCS1 is one of the major functional targets of XIAP in Treg cells and that re-introduction of SOCS1 corrects the functional defects and plasticity of *Xiap*[−/−] tTreg cells.

**WT Treg cells rescue *Xiap*[−/−] mice from inflammatory death**. It has previously been demonstrated that *Xiap*-deficient mice are highly susceptible to infection by selective pathogens such as *C. albicans*[12–15], with resulting syndromes analogous to XLP-2. To determine whether impaired *Xiap*[−/−] Treg cells activity contributed to the sensitivity of *Xiap*[−/−] mice to infections, we analyzed tTreg cells isolated from mice infected with *C. albicans*.

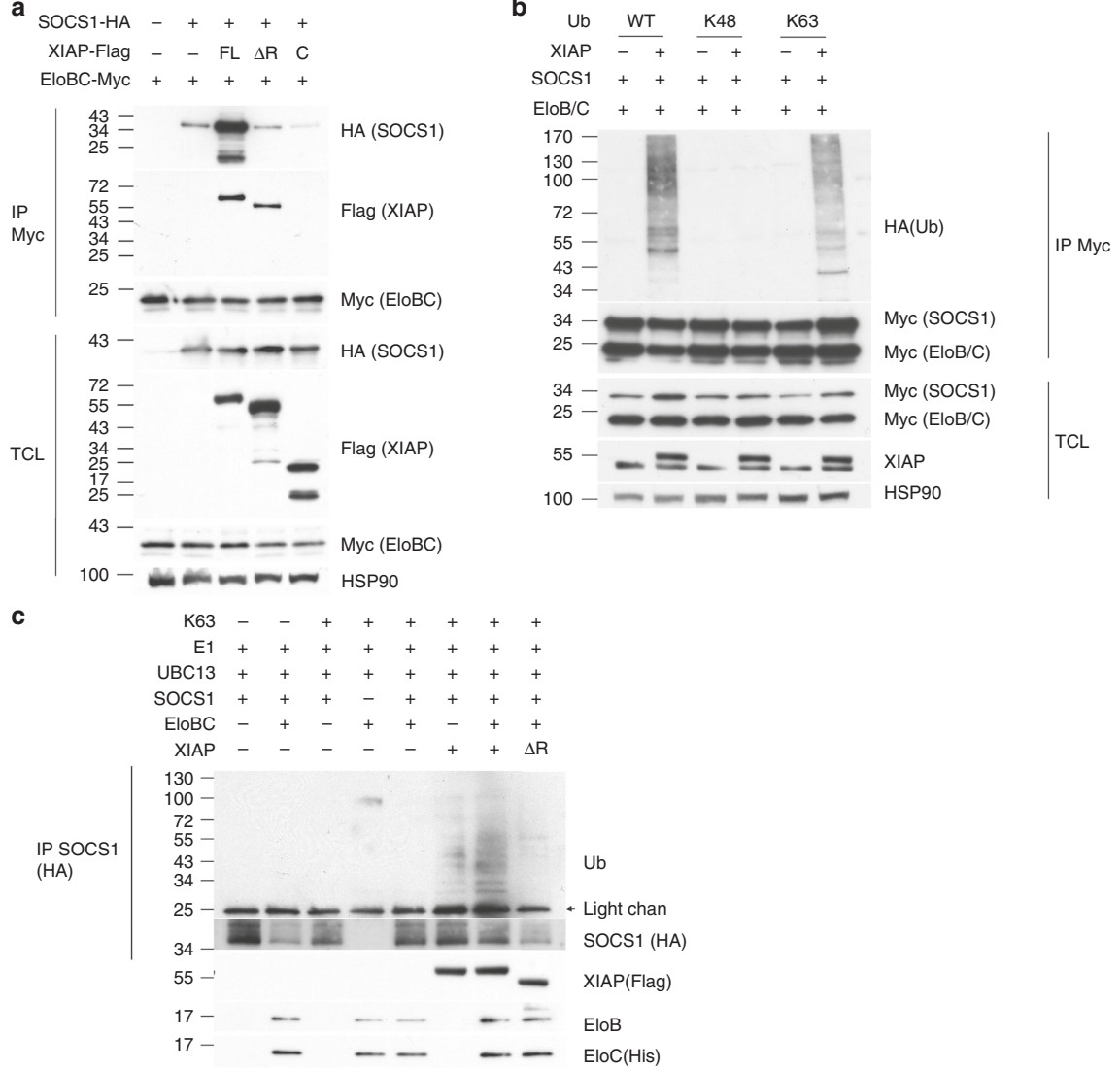

**Fig. 4** XIAP enhances the association of Elongin B/C with SOCS1 and promotes SOCS1 K63 ubiquitination. **a** XIAP increases the association of Elongin B/C with SOCS1. Full-length, ΔRING (ΔR) or C-terminal (C) XIAP-FLAG was co-transfected with SOCS1-HA and Elongin B/C-Myc into HEK293T cells. Elongin B/C-Myc in cell lysates was pulled down by anti-Myc and the presence of SOCS1 and XIAP in the precipitates and lysates was determined by the respective antibodies. **b** XIAP enhances the addition of K63 ubiquitin to SOCS1 in vivo. XIAP-FLAG was co-transfected with SOCS1-myc and Elongin B/C-Myc into HEK293T cells, with WT Ub, K48 Ub or K63 Ub, as indicated. SOCS1 was immune-precipitated, and the association of ubiquitin type determined. **c** XIAP promotes SOCS1 K63 poly-ubiquitination in vitro. Recombinant K63 ubiquitin-HA, E1, E2 (UBC13), SOCS1-HA (on Mag beads), Elongin B/C-Myc, or XIAP-FLAG was added as indicated to a ubiquitination reaction conducted at 30 °C for 1 h. Around 10% of the reaction mixture was used for Western blotting. SOCS1-HA was pulled down from the rest of the reaction mixture and ubiquitination of the SOCS1 complex was determined. Experiments were independently repeated three times (**a**, **c**) or twice (**b**)

Low-dose ($1 \times 10^5$) *C. albicans* infection killed $Xiap^{-/-}$ mice after 10 days, but did not affect the viability of most WT mice[15]. We isolated Treg cells from WT and $Xiap^{-/-}$ mice 10 days after *C. albicans* infection. CD4$^+$Foxp3$^+$ cell frequency was comparable between infected WT and $Xiap^{-/-}$ mice (Fig. 6a). However, the fractions of IFN-γ$^+$Foxp3$^+$ and IL-17A$^+$Foxp3$^+$ Treg cells in $Xiap^{-/-}$ mice were higher than those in WT mice (Fig. 6a), suggesting an increased conversion of $Xiap^{-/-}$ Treg cells to IFN-γ$^+$- and IL-17-secreting cells after *C. albicans* infection.

To determine whether impaired Treg cells activity contributed to the sensitivity of $Xiap^{-/-}$ mice to infections, WT iTreg cells were transferred into $Xiap^{-/-}$ mice after *C. albicans* infection. Administration of WT iTreg cells 2 days after low-dose *C. albicans* infection partly rescued the survival of

$Xiap^{-/-}$ mice; 60% of infected $Xiap^{-/-}$ mice that received WT iTreg cells survived 40 days after infection, whereas all untreated $Xiap^{-/-}$ mice had died by 18 days post-infection (Fig. 6b). WT iTreg cells transfer also alleviated kidney inflammation in infected $Xiap^{-/-}$ mice, as demonstrated by diminished kidney neutrophil infiltration (Fig. 6c). In addition, the elevated levels of serum inflammatory cytokines in untreated infected $Xiap^{-/-}$ mice were profoundly suppressed by adoptive transfer of WT iTreg cells (Fig. 6d). These results suggest that deficient inflammation control in $Xiap^{-/-}$ mice is partly due to impaired Treg cells functioning and that XIAP-intact iTreg cells restore the ability of $Xiap^{-/-}$ mice to respond to infection-induced inflammation.

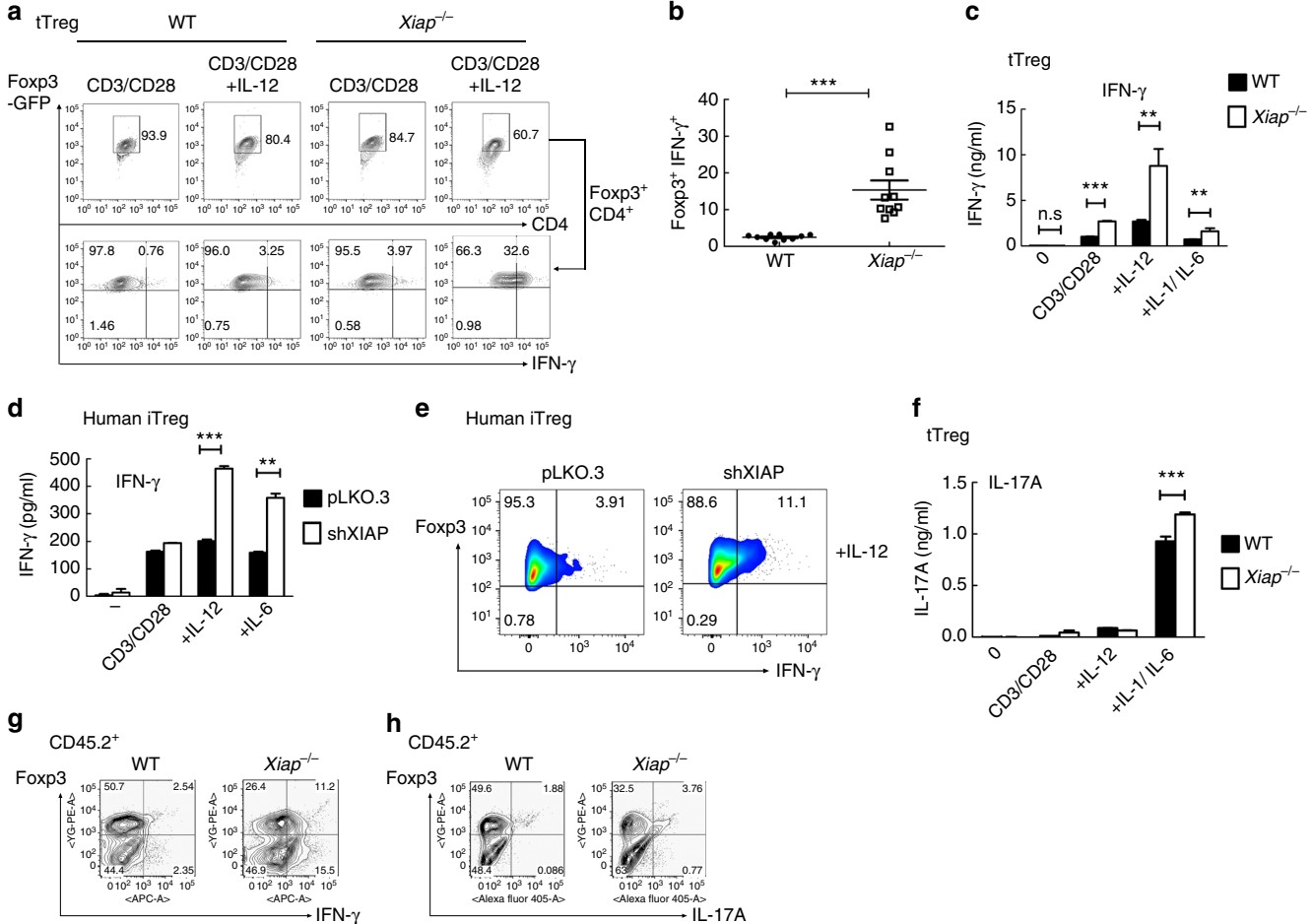

**Fig. 5** Enhanced conversion of XIAP-deficient Treg cells into Foxp3+IFN-γ+ cells. **a**, **b** Increased transition of $Xiap^{-/-}$ tTreg cells into IFN-γ-producing cells. WT and $Xiap^{-/-}$ tTreg cells (Foxp3-GFP tagged) were isolated by GFP-expression and re-stimulated with plate-bound anti-CD3/CD28 and IL-2 (CD3/CD28), with additional IL-12 ( + IL-12, 50 ng ml$^{-1}$), for 4 days. tTreg cells were then reactivated with TPA/A23187 for 6 h, and the expressions of IFN-γ in the gated GFP+ (representing Foxp3+) fraction were determined by intracellular staining (**a**). Quantitation of Foxp3+IFN-γ+ cells from tTreg cells isolated by either Foxp3-GFP or CD4+CD25+ expression (**b**), $n = 10$. ***$P < 0.001$ for unpaired $t$-test. **c** Increased secretion of IFN-γ by $Xiap^{-/-}$ tTreg cells. Purified WT and $Xiap^{-/-}$ tTreg cells were stimulated as in (**a**), or with additional IL-6 plus IL-1α + IL-1β ( + IL-6/IL-1), and re-activated with TPA/A23187 for 24 h and the levels of IFN-γ in supernatants was analyzed by ELISA. **d**, **e** Enhanced expression and secretion of IFN-γ by human XIAP-knockdown iTreg cells. Human control and XIAP-knockdown iTreg cells were activated by anti-CD3/CD28 and IL-2, with the addition of IL-12 or IL-6, as indicated, for 4 days. Secretion of IFN-γ was quantitated (**d**) and intracellular expressions of Foxp3 and IFN-γ were determined (**e**). **f** Elevated secretion of IL-17A by $Xiap^{-/-}$ tTreg cells. WT and $Xiap^{-/-}$ tTreg cells were stimulated as in (**c**) and the generated IL-17 was quantitated by ELISA. **g**, **h** Increased expression of IFN-γ and IL-17 in the adoptively transferred $Xiap^{-/-}$ tTreg cells. WT or $Xiap^{-/-}$ CD45.2+ tTreg cells were administrated into $Rag1^{-/-}$ CD45.1+ mice together with WT CD45.1+ CD4+CD25− effector T cells. Mice were killed after 27 days and CD4+ T cells from spleen and lymph nodes were isolated. CD4+ T cells were activated by TPA/A23187 and the expressions of Foxp3, IFN-γ (**g**) and IL-17A (**h**) in CD45.2+ cells were determined by intracellular staining. Values (**c**, **e**, **f**) are mean ± SD of triplicate samples in an experiment. *$P < 0.05$, **$P < 0.01$, ***$P < 0.001$ for unpaired $t$-test. All experiments (**a**–**h**) were independently repeated three times with similar results

**Anti-IL-6R rescues defects in $Xiap^{-/-}$ Treg cells.** Among the serum inflammatory cytokines analyzed in *C. albicans*-infected $Xiap^{-/-}$ mice, the high levels of IL-6 were particularly noticeable (Fig. 6d). Therefore, we examined whether the impaired functioning of $Xiap^{-/-}$ Treg cells was associated with IL-6. IL-6 is generated by many different types of cells. Co-stimulation with IL-12 also induced production of IL-6 from human XIAP-deficient iTreg cells (Supplementary Figure 10). Inclusion of anti-IL-6R effectively suppressed TCR/CD28-induced IFN-γ production in $Xiap^{-/-}$ iTreg cells with or without IL-12 (Fig. 7a), and reduced the fraction of IFN-γ-expressing $Xiap^{-/-}$ iTreg cells activated through TCR and IL-12 (Fig. 7b). The effectiveness of IL-6R blockage in inhibiting conversion into IFN-γ+ tTreg cells was also confirmed in Foxp3-GFP-tagged $Xiap^{-/-}$ tTreg cells (Supplementary Figure 11). In addition, inclusion of anti-IL-6R decreased the expression and secretion of IL-6 in $Xiap^{-/-}$ Treg cells induced by IL-12 and/or TCR/CD28 (Supplementary Figure 12). Inclusion of anti-IL-6R also inhibited the expression and secretion of IFN-γ in human control and XIAP-knockdown iTreg cells (Fig. 7c and d). The production of IL-17 and IL-6 in human control and XIAP-deficient iTreg cells were similarly suppressed by anti-IL-6R (Supplementary Figure 13). Thus, anti-IL-6R effectively suppresses IFN-γ, IL-6, or IL-17 generation in XIAP-deficient Treg cells stimulated by TCR/CD28 with IL-12 or IL-6.

We also examined whether blocking other inflammatory cytokines, such as TNF or IL-1β, reversed the plasticity of $Xiap^{-/-}$ iTreg cells. Both anti-TNF and anti-IL-1R failed to antagonize activation- or IL-12-induced IFN-γ secretion in $Xiap$

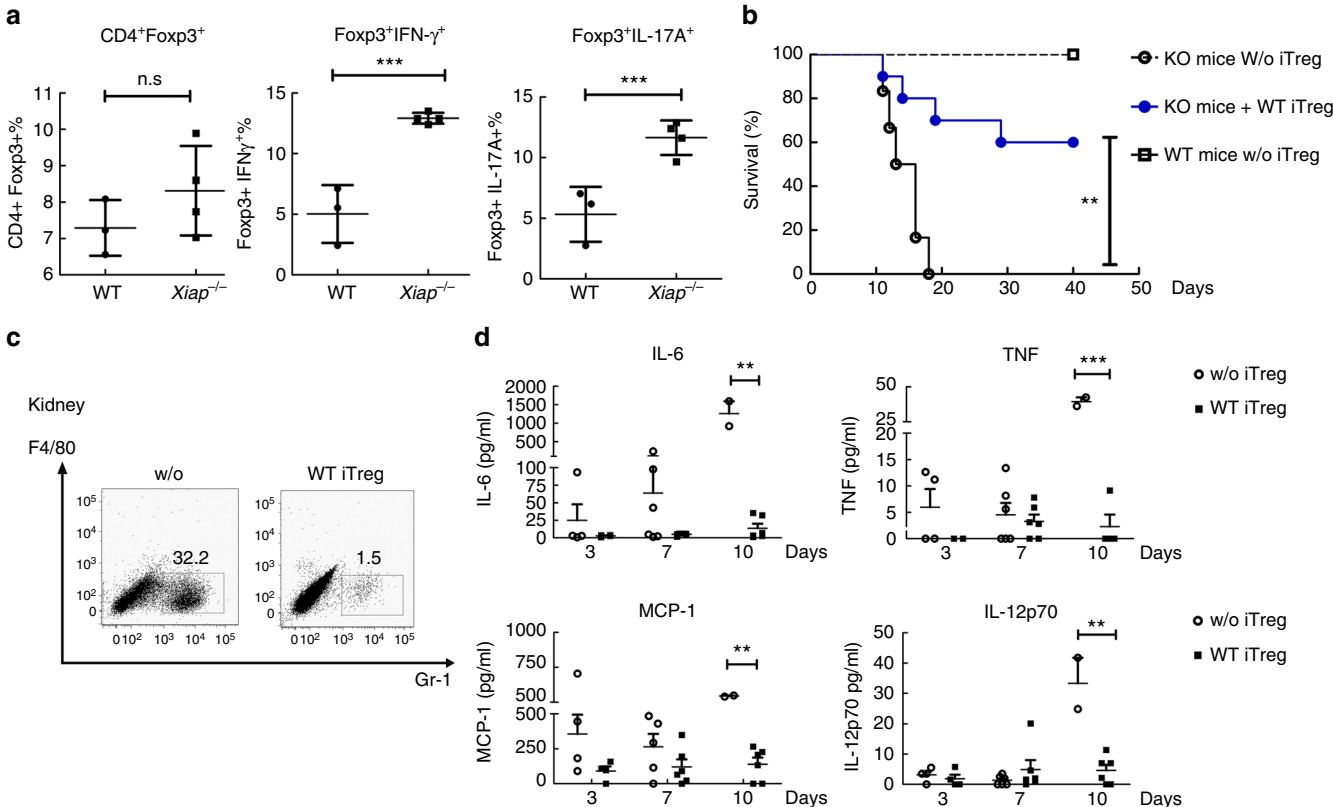

**Fig. 6** Transfer of Treg cells rescues $Xiap^{-/-}$ mice from lethal *Candida albicans* infection. **a** Increased expression of IFN-γ[+] and IL-17[+] cells in $Xiap^{-/-}$ Treg cells upon *Candida albicans* infection. WT and $Xiap^{-/-}$ female mice were intravenously infected with *C. albicans* ($1 \times 10^5$), and mice were killed at day 10. T cells were isolated from spleen, and the frequencies of CD4[+]Foxp3[+], Foxp3[+]IFN-γ[+] and Foxp3[+]IL-17[+] cells were quantitated by intracellular staining. **b** Transfer of WT iTreg cells partially rescues $Xiap^{-/-}$ mice from *C. albicans*-induced lethality. WT and $Xiap^{-/-}$ female mice were intravenously administered with *C. albicans* ($1 \times 10^5$) and a group of $Xiap^{-/-}$ mice also received WT iTreg cells ($1 \times 10^6$) at day 2. Survival of mice was monitored and is presented as a Kaplan–Meier survival curve ($n = 7$ for each group). **$P < 0.01$. **c** Reduced kidney inflammation in *C. albicans*-infected $Xiap^{-/-}$ mice with WT iTreg cells transfer. Kidney was isolated from $Xiap^{-/-}$ mice 10 days after *C. albicans* injection and infiltrated neutrophil contents were determined. **d** Reduced inflammatory cytokine production in *C. albicans*-infected $Xiap^{-/-}$ mice into which WT iTreg cells had been transferred. Serum from mice was collected at the indicated time-points after *C. albicans* injection, and the levels of IL-6, TNF, MCP-1, and IL-12 were determined, $n = 6$. Values (**a**, **d**) are mean ± SD of samples. *$P < 0.05$, **$P < 0.01$, ***$P < 0.001$ for unpaired *t*-test

$Xiap^{-/-}$ Treg cells (Fig. 7e). Anti-TNF did not prevent IL-12-costimulated IFN-γ expression in $Xiap^{-/-}$ iTreg cells (Supplementary Figure 14). In addition, anti-IL-1R increased IFN-γ expression in $Xiap^{-/-}$ iTreg cells co-stimulated with IL-12 (Supplementary Figure 14). Furthermore, inclusion of anti-IFN-γ did not effectively prevent IL-12-stimulated generation of IFN-γ in $Xiap^{-/-}$ iTreg cells (Supplementary Figure 15a), and exhibited no effect on the production of IL-17 promoted by IL-1 and IL-6 (Supplementary Figure 15b). Of the four anti-inflammatory antibodies we examined, anti-IL-6R was the most effective in preventing production of IFN-γ in XIAP-deficient Treg cells.

The effectiveness of anti-IL-6R in correcting defective $Xiap^{-/-}$ tTreg cells was also examined in vivo. Effector T cell-induced colitis in $Rag1^{-/-}$ mice, which was poorly rescued by $Xiap^{-/-}$ tTreg cells (Fig. 1g), was prevented by co-administration of anti-IL-6R (Fig. 7f and Supplementary Figure 16). The reduction in the Foxp3[+] cell fraction of transferred CD45.2[+] $Xiap^{-/-}$ tTreg cells (Fig. 2c and d) was fully restored by co-injection of anti-IL-6R into $Rag1^{-/-}$ mice (Fig. 7g). In addition, the appearance of IFN-γ[+]Foxp3[+] cells in CD45.2[+] $Xiap^{-/-}$ tTreg cells was suppressed by co-administration of anti-IL-6R into $Rag1^{-/-}$ mice (Fig. 7h). Thus, the plasticity of $Xiap^{-/-}$ tTreg cells in vitro and in vivo can be prevented by blockage of IL-6R.

**Anti-IL-6R restores the function of $Xiap^{-/-}$ Treg cells**. We further examined whether a combination of anti-IL-6R and $Xiap^{-/-}$ Treg cells restored the resistance of $Xiap^{-/-}$ mice to infection-induced inflammation and lethality. $Xiap^{-/-}$ mice were infected with a lethal dose of *C. albicans* that did not affect WT mice. $Xiap^{-/-}$ iTreg cells were administered 2 days after infection, with or without by anti-IL-6R. Either $Xiap^{-/-}$ iTreg cells or anti-IL-6R partly increased the survival of $Xiap^{-/-}$ mice, though the increase was not statistically significant (Fig. 8a). Combinatory treatment of anti-IL-6R and $Xiap^{-/-}$ iTreg cells effectively rescued $Xiap^{-/-}$ mice from inflammatory infection (Fig. 8a). The elevated serum IL-6 levels in untreated infected $Xiap^{-/-}$ mice were suppressed by the combination of $Xiap^{-/-}$ iTreg cells and anti-IL-6R (Fig. 8b). We further traced the fate of CD45.1[+] $Xiap^{-/-}$ iTreg cells after their transfer into infected CD45.2[+] $Xiap^{-/-}$ mice. CD45.1[+] $Xiap^{-/-}$ iTreg cells became inflammatory, with abundant IFN-γ[+] expression in $Xiap^{-/-}$ mice pre-infected with *C. albicans* (Fig. 8c and d). The administration of anti-IL-6R inhibited the conversion of CD45.1[+] $Xiap^{-/-}$ Treg cells into IFN-γ[+]Foxp3[+] cells (Fig. 8c and d). The additive therapeutic effect of anti-IL-6R application is demonstrated by the fact that a combination of anti-IL-6R and $Xiap^{-/-}$ Treg cells reduced kidney neutrophil infiltration more effectively than either treatment alone (Fig. 8e). Notably, a significant decrease in kidney fungal

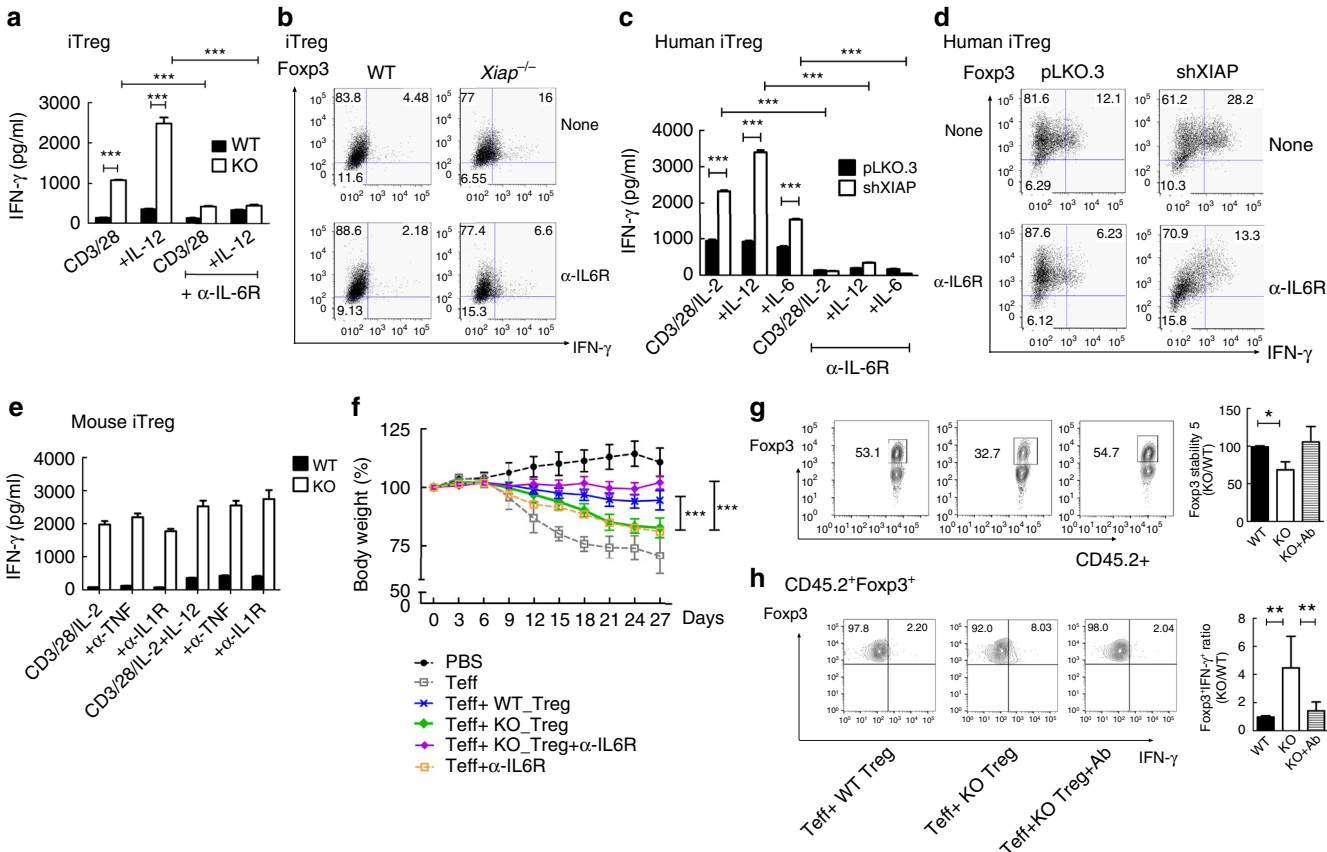

**Fig. 7** Anti-IL-6R reduces the expression of IFN-γ in activated *Xiap*⁻/⁻ Treg cells. **a, b** Anti-IL-6R decreases IFN-γ expression in re-stimulated *Xiap*⁻/⁻ Treg cells. WT and *Xiap*⁻/⁻ iTreg cells were stimulated with anti-CD3/CD28 and IL-2, or with additional IL-12 as indicated, in the absence or presence of anti-IL-6R (50 μg ml⁻¹) for 4 days. iTreg cells were re-stimulated with TPA/A23187 for 24 h and IFN-γ production was determined by ELISA (**a**), or reactivated with TPA/A23187 for 5 h and the expressions of Foxp3 and IFN-γ were determined by intracellular staining (**b**). **c, d** Anti-IL-6R inhibits IFN-γ expression in human Treg cells. Control and human XIAP-knockdown iTreg cells were stimulated as in (**a, b**), with additional IL-6 as indicated, and secretion (**c**) or intracellular expression (**d**) of IFN-γ was determined. **e** Inability of anti-TNF or anti-IL-1R to inhibit the production of IFN-γ in activated *Xiap*⁻/⁻ Treg cells. WT and *Xiap*⁻/⁻ iTreg cells were stimulated, as described in (**a**), in the presence or absence of anti-TNF or anti-IL-1R (50 μg ml⁻¹ each) for 4 days. Treg cells were re-stimulated with TPA/A23187 for 24 h and the secreted IFN-γ was determined by ELISA. **f** Anti-IL-6R rescues the impaired suppressive activity of *Xiap*⁻/⁻ tTreg cells in vivo. CD45.2⁺ WT or *Xiap*⁻/⁻ tTreg cells were co-transferred with CD45.1⁺ CD4⁺CD25⁻ effector T cells into male CD45.1⁺ *Rag1*⁻/⁻ mice. Anti-IL-6R antibody (500 μg per mouse) was intraperitoneally administrated at day 0, followed by weekly dosing of 500 μg. The body weights of mice were monitored at the indicated time-points. ***$P < 0.001$ by two-way ANOVA. **g, h** Anti-IL-6R restores Foxp3 stability and reduces IFN-γ expression in *Xiap*⁻/⁻ tTreg cells transferred in vivo. Lymphocytes were isolated from spleen and mesenteric lymph nodes of mice in (**f**) and reactivated with TPA/A23187. The expressions of Foxp3 (**g**) and IFN-γ (**h**) in CD45.2⁺ T cells were determined by intracellular staining. Values are mean ± SD of triplicate samples in an experiment. *$P < 0.05$, **$P < 0.01$, ***$P < 0.001$ for unpaired *t*-test (**a, c, g, h**). Experiments (**a–e**) were independently repeated three times with similar results

titers was detected for *Xiap*⁻/⁻ Treg cells treatment, with or without anti-IL-6R (Fig. 8f). We further investigated whether the presence of anti-IL-6R dampened the T helper 17 responses, since Th17 plays a prominent role in resolving *C. albicans* infection[46–48]. Figure 8g illustrates that IL-17 production by T cells isolated from infected mice was not affected by anti-IL-6R or anti-IL-6R plus *Xiap*⁻/⁻ Treg cells. These results suggest that inhibition of Treg cells re-programming by anti-IL-6R significantly increased the functioning of XIAP-deficient Treg cells in vivo. Moreover, we have illustrated, likely for the first time, that in conjunction with anti-IL-6R, functionally unstable Treg cells could be used to treat inflammatory diseases.

## Discussion

XIAP is a caspase-binding anti-apoptotic protein critical for cell survival. Although XIAP-knockout did not affect Treg cells development, the expression of Treg cells markers or the production by Treg cells of IL-10 and TGF-β, XIAP-deficient Treg cells were defective in their suppressive activity both in vitro and in vivo (Fig. 1). We found that SOCS1 was specifically reduced in XIAP-deficient T cells (Fig. 3). We also demonstrate that XIAP binds to SOCS1 and increases the protein stability of SOCS1, likely by promoting its K63 ubiquitination (Fig. 4). In contrast, K48 ubiquitination of SOCS1 was not affected by XIAP. Since SOCS1 is critical to maintaining the inhibitory activity of Treg cells[36], XIAP-deficient Treg cells exhibited a compromised suppressive ability. We further demonstrate that re-introduction of SOCS1 corrects defects of *Xiap*⁻/⁻ Treg cells (Supplementary Figure 9). Thus, the impaired Treg cells functioning in *Xiap*⁻/⁻ Treg cells could be partly attributed to a downregulation of SOCS1. Our results reveal an unexpected requirement for XIAP in Treg cells.

We have identified two different defects that account for the diminished suppressive activity in *Xiap*⁻/⁻ Treg cells. The stability

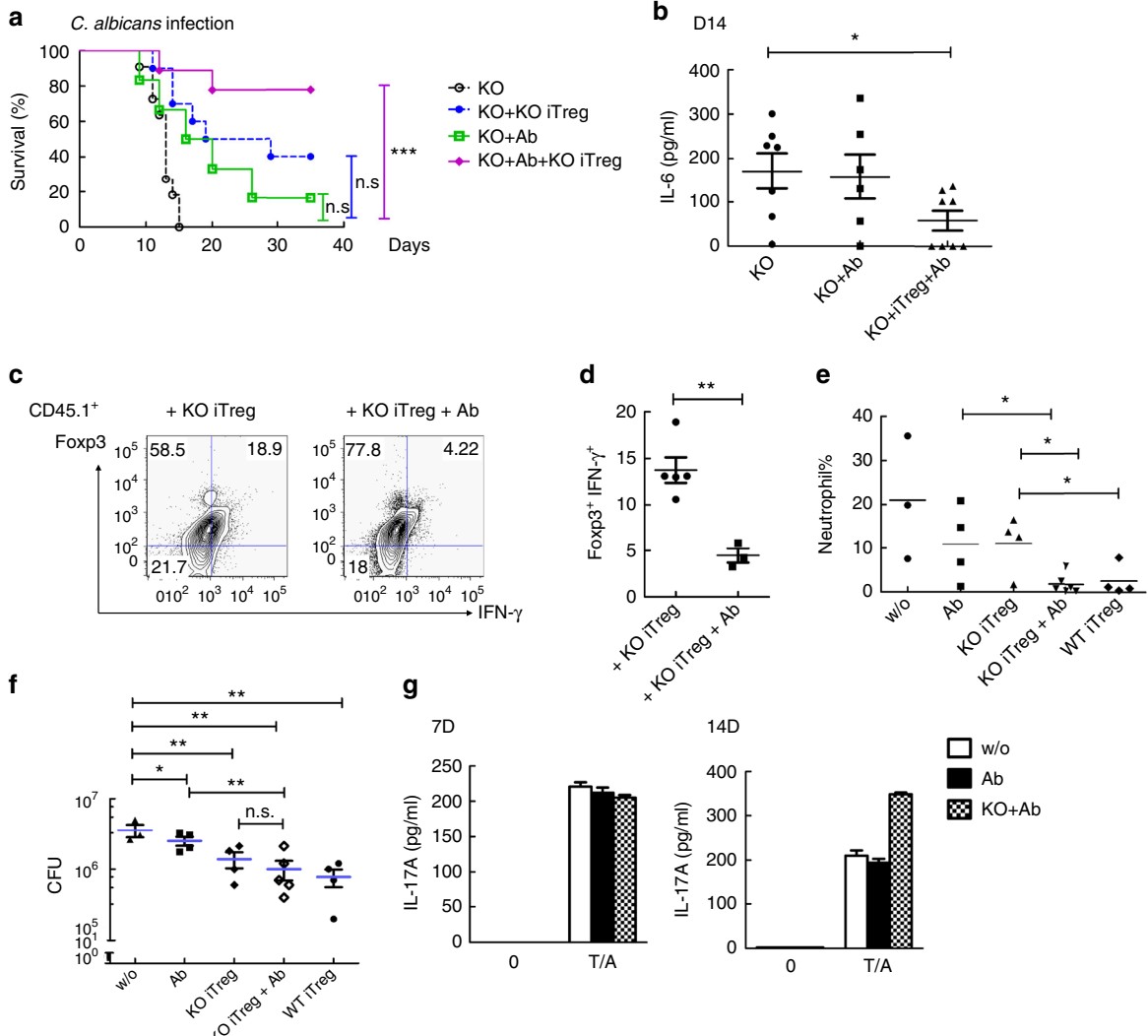

**Fig. 8** Combination anti-IL-6R and *Xiap*$^{-/-}$ Treg cells rescue mice from infection-induced inflammation. **a**, **b** *Xiap*$^{-/-}$ iTreg cells plus anti-IL-6R rescue *Xiap*$^{-/-}$ mice from *C. albicans* infection. Male *Xiap*$^{-/-}$ mice (KO) were intravenously administered with *C. albicans* and *Xiap*$^{-/-}$ iTreg cells ($1 \times 10^6$) or anti-IL-6R antibody (500 µg) at day 2, or both, as indicated. Survival of mice is presented as a Kaplan–Meier survival curve (**a**). $n = 11$ for KO, $n = 10$ for KO + KO iTreg, $n = 6$ for KO + Ab, $n = 9$ for KO + KO iTreg + Ab. ***$P < 0.001$ for Log-rank (Mantel–Cox) Test (**a**). n.s., not significant. The serum level of IL-6 was determined on day 14 after infection (**b**). *$P < 0.05$ for unpaired *t*-test (**b**). **c**, **d** Anti-IL-6R prevents the conversion of the transferred *Xiap*$^{-/-}$ Treg cells. CD45.1$^+$ *Xiap*$^{-/-}$ iTreg cells were transferred into CD45.2$^+$ *Xiap*$^{-/-}$ mice infected with *C. albicans* as in (**a**). CD45.1$^+$ T cells were recovered at day 10 post-infection, and the expression of Foxp3 and IFN-γ was determined (**c**). The Foxp3$^+$IFN-γ$^+$ fractions in the CD45.1$^+$ Treg cells were quantitated (**d**), $n = 5$. **$P < 0.01$ for unpaired *t*-test (**d**). **e**, **f** Anti-IL-6R reduces kidney neutrophil infiltration and fungal load in *C. albicans*-infected *Xiap*$^{-/-}$ mice with KO iTreg cells transfer. Male *Xiap*$^{-/-}$ mice were infected with *C. albicans*, as described in (**a**). Kidneys were isolated 14 days post-infection and neutrophils (**e**) and *C. albicans* titres (**f**) were quantitated. *$P < 0.05$, **$P < 0.01$ for unpaired *t*-test. n.s., not significant. **g** Anti-IL-6R does not affect IL-17 production. Male CD45.1$^+$ *Xiap*$^{-/-}$ mice were infected with *C. albicans* and treated with anti-IL-6R and anti-IL-6R plus CD45.2$^+$ KO iTreg cells. Spleen T cells and B cells were isolated 7 days after treatments. CD45.2$^+$ iTreg cells were depleted. T cells were incubated with B cells and heat-killed *C. albicans* (HKCA) for 4 days, and the production of IL-17A after stimulation with TPA/A23187 (10/100 ng ml$^{-1}$) was determined by ELISA[47, 48]. Values are mean ± SD of triplicate samples in an experiment. Experiments were independently repeated three times with similar results

of Foxp3 protein was reduced in activated *Xiap*$^{-/-}$ Treg cells (Fig. 2). Since persistent expression of Foxp3 is required to maintain the suppressive activities of Treg cells[26, 27], the impaired functioning of *Xiap*$^{-/-}$ Treg cells may be partly attributable to decreased Foxp3 presence. Notably, even though IL-2 is involved in Foxp3 stabilization[49, 50], the reduced Foxp3 stability in *Xiap*$^{-/-}$ Treg cells is IL-2-independent. IL-2-induced STAT5 activation was enhanced in *Xiap*$^{-/-}$ Treg cells (Supplementary Figure 8), similar to those in *Socs1*$^{-/-}$ T cells[51].

A more profound defect of XIAP-deficient Treg cells was the increased conversion into IFN-γ-, IL-6-, and IL-17-producing Treg cells (Fig. 5). IL-12 is known to suppress Treg cells and converts Treg cells into Th1-type cells[52]. Deletion of SOCS1 elicits excess activation of the IFN-γ-STAT1 axis[36, 37]. Consistent with these observations, we found that IL-12 enhances WT and XIAP-deficient Treg cells to produce IFN-γ (Fig. 5). Previous studies have shown that pathogenic Treg cells lose Foxp3 expression and produce IFN-γ[28, 53]. IFN-γ promotes tissue inflammation, and IFN-γ-expressing *Xiap*$^{-/-}$ Treg cells likely

contribute to inflammatory pathology. XIAP-deficiency also conferred on Treg cells an enhanced production of IL-6 and IL-17 (Fig. 5f, Supplementary Figures 10, 12, 13). The increase in IFN-γ generation is more pronounced than for enhancement of IL-17 expression in XIAP-deficient Treg cells in vitro (Fig. 5c vs. 5f, Fig. 5d vs. Supplementary Figure 13b) and in vivo (Fig. 5g and h), suggesting that XIAP-deficiency preferentially re-programs the development of Treg cells into Th1-like cells. IL-2 inhibits Th17 differentiation[54], whether enhanced IL-2 responses in $Xiap^{-/-}$ Treg cells contributes to preferential Th1 over Th17 reprogramming is currently being investigated.

The critical role of SOCS1 in maintaining Treg stability has been demonstrated[36, 55]. In the present study, the defective activity of $Xiap^{-/-}$ Treg cells could be partly attributed to a loss of SOCS1. Overexpression of SOCS1 in $Xiap^{-/-}$ Treg cells prevented their conversion into IFN-γ-expressing Treg cells and restored the stability of Foxp3 (Supplemental Figure 9). We also demonstrate that XIAP increases the stability of SOCS1. SOCS1 is an E3 ligase known to target signaling molecules, such as VAV1 and Jak2, for ubiquitination and degradation. Our results indicate that, in the absence of XIAP, SOCS1 protein itself is subjected to degradation. Together with the capacity of XIAP to promote SOCS1 K63 polyubiquitination, our results suggest that ubiquitination of SOCS1 likely represents a new level of regulation responsible for the stability and function of the SOCS1 E3 ligase.

Despite the similar susceptibility between $Socs1^{f/f} Lck^{Cre}$ Treg cells[36] and $Xiap^{-/-}$ Treg cells to generate IFN-γ and their defective abilities to inhibit colitis (Fig. 1g and h), Treg-specific-knockout of SOCS1 ($Socs1^{f/f} Foxp3^{Cre}$) generates Treg cells that can still suppress colitis in $Rag2^{-/-}$ mice[55]. It has been suggested that environmental inflammatory cues prime $Socs1^{-/-}$ Treg cells to become IFN-γ-producing cells[55]. We have previously shown that myeloid components in $Xiap^{-/-}$ mice become inflammatory upon infection[15], suggesting a possibility that $Xiap^{-/-}$ myeloid cells prime $Xiap^{-/-}$ Treg cells for plasticity. We plan to investigate whether the conversion of $Xiap^{-/-}$ Treg cells into inflammatory ex-Treg cells involves conditioning from $Xiap^{-/-}$ myeloid cells using mice with Treg-specific knockout of XIAP.

Our results also illustrate that defects in XIAP-deficient Treg cells are manifested by inflammatory cytokines. TCR/CD28 activation induces low-level production of IFN-γ, IL-6 and IL-17 in XIAP-deficient Treg cells (Fig. 5 and Supplementary Figures 10–13). Only upon co-stimulation with IL-12 or IL-1/IL-6 does the expression of IFN-γ, IL-6 or IL-17 become prominent in XIAP-deficient Treg cells. Therefore, Treg cells do not manifest pathogenic consequences in $Xiap^{-/-}$ mice before any inflammatory stimulation. We found that elevated serum IL-6 was the most prominent among the several inflammatory cytokines we measured upon C. albicans lethal infection in $Xiap^{-/-}$ mice (Fig. 6). IL-6 affects the stability of Treg cells[56], and blockage of IL-6 increases the population of Treg cells[57, 58]. We further demonstrate that anti-IL-6R effectively inhibited IL-12-induced production of IFN-γ, while maintaining Foxp3 expression in $Xiap^{-/-}$ Treg cells (Fig. 7). Surprisingly, other anti-inflammatory biologics, i.e., anti-TNF, anti-IL-1R, or anti-IFN-γ did not effectively prevent IFN-γ generation in $Xiap^{-/-}$ Treg cells (Fig. 7e, Supplementary Figures 14 and 15a), illustrating the distinct role of IL-6R in $Xiap^{-/-}$ Treg cells conversion. Anti-IL-6R would be expected to prevent conversion of Treg cells into Th17-like cells, but our results illustrate that anti-IL-6R is unique in maintaining Treg cells stability in a Th1-prone environment.

Among the possible complications of anti-IL-6R is the increased susceptibility to infection and reduced generation of Th17 cells. In the present experiment, anti-IL-6R was administered 2 days after C. albicans infection. Notably, anti-IL-6R decreased inflammation, evidenced by both inflammatory

cytokine levels and kidney neutrophil infiltration (Fig. 8b and e). Fungal load was actually reduced in mice treated with anti-IL-6R or anti-IL-6R plus $Xiap^{-/-}$ iTreg cells (Fig. 8f). In addition, anti-IL-6R did not affect antifungal Th17 responses (Fig. 8g). We speculate that a single dose of anti-IL-6R acts to prevent the conversion of $Xiap^{-/-}$ Treg cells and to attenuate inflammation, but it is not sufficient to attenuate antifungal immunity nor to suppress Th17 responses.

XIAP-deficiency leads to XLP-2 in patients, who exhibit over-activation of macrophages and lymphocytes. $Xiap^{-/-}$ mice succumb to infection by various pathogens[12–15]. Our results and those of others suggest that XIAP-deficient individuals are unable to clear infection due to primary immunodeficiency and persistent inflammation, leading to XLP-2 and lethality. As a primary immunodeficiency disease, the excess activation of lymphocytes in XLP-2 is likely a consequence of myeloid cell over-activation. Here, we have identified a subtle defect in the adaptive immunity of $Xiap^{-/-}$ mice related to their Treg cells. $Xiap^{-/-}$ Treg cells were mostly normal before infection, but were converted into IFN-γ-, IL-6-, and IL-17-expressing T cells in an inflammatory environment. Normal Treg cells are known to inhibit various immune cells[59]. Our observation that transfer of WT Treg cells effectively prevented infection-induced inflammation and conferred survival after an otherwise lethal infection in $Xiap^{-/-}$ mice (Fig. 6b–d) supports the notion that Treg cells are defective in $Xiap^{-/-}$ mice and XLP-2 patients, and that correction of XIAP-deficiency restores functional Treg cells. Our findings may also explain how lymphocytes become over-activated in XLP-2 patients, as Treg cells are required to keep lymphocyte activation in check. We propose a scenario whereby $Xiap^{-/-}$ mice are unable to control early infection as a consequence of impaired innate immunity, and inflammation caused by persistent pathogen presence primes for aberrant inflammatory Treg cells activation, leading to further escalated lymphocyte activation in $Xiap^{-/-}$ mice.

As a primary immunodeficiency disease, the only curative treatment for XLP-2 patients is allogeneic hematopoietic cell transplantation (HCT) to restore expression of XIAP in hematopoietic stem cells (HSCs), yet outcomes are limited by the toxicity associated with transplantation[60]. Even though transduction of the WT gene into hematopoietic progenitor cells from patients would in theory rescue the genetic defect[61], autologous HSC transplantation in XLP-2 patients is difficult given the treatments required to eradicate endogenous hematopoietic progenitor cells in these ill pediatric patients. As an alternative, adoptive T cell immunotherapy has been explored for treatment of primary immunodeficiency diseases linked to viral infection[62]. Our study illustrates another possibility of using Treg cells in the treatment of inflammatory diseases caused by innate immunodeficiency. Our results (Fig. 6) suggest that administration of XIAP-reconstituted Treg cells could help reduce the excess immuno-related inflammation seen in XLP-2 patients. XIAP could be re-introduced into T cells from XLP-2 patients, and these XIAP-restored T cells could then be differentiated into Treg cells in vitro in sufficient quantities. Expression of XIAP in T cells could thus be considered an improvement over transduction of XIAP in HSCs from XLP-2 patients. It may be noted that, due to the low proliferative ability of mouse tTreg cells, we used mouse iTreg cells to test such a possibility (Fig. 6). Given that tTreg cells are more stable than iTreg cells, human tTreg cells are expected to be more effective than the iTreg cells shown in Fig. 6 in suppressing infection-induced inflammation.

We further advanced our therapeutic effect in mice by using a combination of anti-IL-6R and XIAP-deficient Treg cells. We illustrate that anti-IL-6R prevents the re-programming of $Xiap^{-/-}$ Treg cells into inflammatory Treg cells, and that $Xiap^{-/-}$ Treg cells become effective inhibitors of inflammation in the presence

of anti-IL-6R (Fig. 8). Inflammation-induced Treg cells re-programming is a major cause of Treg cells inactivation in vivo. Our results suggest that even if Treg cells functioning is substantially impaired by primary mutation, suppression of inflammation by co-administration of anti-IL-6R effectively restores the functioning of Treg cells. Therefore, a combination treatment of anti-IL-6R with autologous Treg cells should be effective in treating the inflammation and pathology of primary immuno-deficiency diseases, bypassing the need to re-introduce the respective WT version of the defective gene into Treg cells.

A recent study revealed that innate immunodeficiency could be rescued by adaptive immunity, evidencing that an impaired response to Staphylococcus infection caused by TLR2 adapter deficiency could be rescued by antibodies against staphylococcal lipoteichoic acid[63]. Our results may be viewed as another way of reversing innate immunodeficiency by adaptive immunity using Treg cells and anti-IL-6R. In our case, anti-IL-6R blocks the Treg cells-destabilizing action of IL-6, allowing full execution of the suppressive effect of functional Treg cells on immuno-related inflammation.

In summary, we have identified a specific defect in XIAP-deficient Treg cells that contributes to the pathogenesis of XLP-2. We further used $Xiap^{-/-}$ mice to demonstrate that XLP-2 could be treated through combinatory use of ex vivo-expanded $Xiap^{-/-}$ iTreg cells and anti-IL-6R. Our results further suggest, most likely for the first time, the possibility of treating primary immunode-ficiency diseases and inflammatory diseases by simultaneous use of autologous Treg cells and anti-IL-6R. Further characterizations may help establish the protocol for clinical applications.

## Methods
**Antibodies and reagents.** Antibodies against pSTAT1$^{Tyr701}$ (#9167, 58D6, 1:1000), pSTAT3$^{Tyr705}$ (#9131, 1:1000), pSTAT5$^{Tyr694}$ (#9359, C11C5, 1:1000), STAT1 (#9172, 1:2000), SOCS1 (#3950s, A156, 1:1000), Myc-tag (#2276, 9B11, 1:1000), and Myc-tag-HRP (#2040s, 9B11, 1:2000) were purchased from Cell Signaling (Beverly, MA). Antibodies against GAPDH (sc-32233, 6C5, 1:4000), E, ElonginB (sc-133090, D5, 1:1000) and normal goat IgG (sc-2028) were purchased from Santa Cruz Biotech (Santa Cruz, CA). Anti-Flag (F1804, M2, 1:2000), anti-Flag-HRP (A8592-2MG, M2, 1:8000), anti-HA (H9658, HA-7, 1:1000) and anti-HA-HRP (H6533, HA-7, 1:4000) were obtained from Sigma (St Louis, MO). Anti-ubiquitin (MAB1510, aka 042691s, 1:1000) and anti-Actin (MAB1501, C4, 1:4000) were purchased from Merck Millipore (Billerica, MA). Anti-STAT5β (13-5300, 1:2000) was purchased from Invitrogen (Waltham, MA) and anti-STAT5a (13-3600, 1:2000) was purchased from Zymed (ThermoFisher Scientific). Anti-STAT3 (ARG54150, AG4C8-1C9-H8, 1:2000) was purchased from Arigo (Basiglio, IT). Anti-SOCS1 (ab9870, 1 μg per test for IP) was purchased from Abcam (Cambridge, UK). Anti-His was purchased from LTK (Taoyuan, Taiwan). Anti-CD4-Pacific blue (100428, GK1.5, 1:100), anti-GITR-PE (YGITR765, 120208, 1:100), anti-CD45.1-BV421 (A20, 110732, 1:100), anti-CD45.2-FITC (104, 109806, 1:100), anti-IFN-γ-APC (XMG1.2, 505810, 1:100), anti-F4/80-APC (BM8, 123116,1:100), anti-CD25-PE (PC61, 102008, 1:100), anti-IL-17A-PE (TC11-18H10.1, 506904, 1:100), anti-CD44-FITC (IM7, 103006, 1:100), anti-IL-6R-APC (D7715A7, 1:100), anti-human CD4-FITC (OKT4, 317408, 1:100), anti-human CD45RA-Pacific blue (HI100, 304123, 1:100), and anti-human CD45RO-PE Cy7 (UCHL1, 304230, 1:100) were purchased from BioLegend (San Diego, CA). Anti-Foxp3-APC (FJK-16S, 17-5773-82, 1:100), anti-Foxp3-PE (FJK-16S, 12-5773-82, 1:100), anti-CTLA4-PE (UC10-489, 12-1522-81, 1:100), anti-LAG3-PE (eBioC9B7W, 12-2231-82, 1:100), anti-FR4-PE (eBio12A5, 12-5445-80, 1:100), anti-CD45.1-PE (A20, 12-0453, 1:100), anti-CD45.2-PE (104, 12-0454-82, 1:100), anti-CD62L-APC (MEL-14, 17-0621-83, 1:100), anti-IL-6-eF450 (MP5-20F3, 48-7061-82, 1:100), anti-human FOXP3-PE (PCH101, 12-4776-42, 1:100), anti-human FOXP3-eF450 (PCH101, 48-4776-41, 1:100), anti-human IFNγ-APC (4SB3m 17-7319-41, 1:100), anti-human CD4-APC (OKT4, 17-0048-42, 1:100), anti-human IL-17A-PE (eBio64CAP17, 12-7178-41, 1:100), and anti-human CD14-PE (61D3, 12-0149-42, 1:100) were purchased from eBioscience (San Diego, CA). Anti-XIAP (610717, 1:1000), anti-HSP90 (610419, 1:4000), anti-IFN-γ-BV480 (XMG1.2, 566097, 1:100), anti-Gr-1-PE (RB6-8C5, 553128, 1:100), anti-human CD127-AF647 (HIL-7R-M21, 558598, 1:50), and anti-human CD25-PE (M-A251, 560989, 1:50) were purchased from BD Biosciences (Franklin Lakes, NJ). Anti-mouse IL-6R (BE0047, 15A7), anti-mouse TNF (BE0058, XT3.11), anti-IL-1R (BE0256, JAMA-147) and anti-human CD28 (BE0248, 9.3) were purchased from BioXCell (West Lebanon, NH). Anti-human IL-6R (Ab00737-10.0-BT, rhPM-1) was purchased from AbsoluteAntibody (Oxford, UK). Anti-CD3 (2C11), anti-CD28 (37.51), anti-mouse CD4 (RL172.4) and anti-human CD3 (OKT3) were purified in our laboratory.

ELISA kits for mouse IFN-γ (88-7314-88), mouse-IL-17A (88-7371-88), mouse IL-6 (88-7064-88), mouse IL-10 (88-7105-88), mouse TGF-β (88-8350-88), human IFN-γ (88-7316-88), human IL-17A (88-7176-88), and human IL-6 (88-7066-86) were purchased from eBioScience. LEAF-purified anti-mouse IFN-γ (R4-6A2, 505707) was purchased from BioLegend. Recombinant mouse or human IL-12, IL-1α and IL-1β were purchased from R&D (Minneapolis, MN). Recombinant mouse IL-2, human IL-2 and mouse IL-6 were purchased from eBioScience. Recombinant human IL-6 and human TGF-β were purchased from Peprotech (Rocky Hill, NJ). CFSE (65-0850-84) and Violet-Tag (425101) were purchased from eBioscience and BioLegend. Heat-killed *C. albicans* was obtained from InvivoGen (San Diego, CA). Recombinant human E1, E2 (UBC13), and Ubiquitin K63 were purchased from Boston Biochem (Cambridge, MA). The human XIAP constructs were previously described[64]. Mouse cDNA was isolated from DO11.10 cells by RT-PCR. pRK5-HA-WT Ub, K63 Ub, and K48 Ub were purchased from Addgene (Cambridge, MA). Sequences of the primers used for cloning of Elongin B/C, XIAP fragments and SOCS1 fragments are listed in Supplementary Table 1.

**Mice.** $Xiap^{-/-}$ mice[10] were a generous gift from Dr. David L. Vaux (Walter and Eliza Hall Institute of Medical Research, Parkville, Australia). $Rag1^{-/-}$ mice and NOD/ShiLt-Tg(Foxp3$^{EGFP/Cre}$)1Jbs/J mice (also known as NOD.Foxp3-EGFP/Cre)[65] were obtained from Jackson Laboratories (Bar Harbor, ME). NOD.Foxp3-EGFP/Cre mice were back-crossed to C57BL/6 mice for 12 generations (referred to as Foxp3-GFP). $Xiap^{-/-}$ mice were crossed with Foxp3-GFP to tag Treg cells with GFP. Mice were maintained in the SPF mouse facility of the Institute of Molecular Biology, Academia Sinica. All mouse experiments were conducted with approval from the Institutional Animal Care and Use Committee, Academia Sinica. All mice used in this study were 8–12 weeks old. The same sex (male or female) mice were used in the same experiment, but opposite sex mice could be used in the repeat of the given experiment. No difference was observed between male and female mice in the analyses conducted in this study. Experimental groups were assigned randomly. Five or more mice in each experimental group was planned, but four mice in some experimental groups, that have been examined in previous studies, were used due to the knockout-mice availability. No blinding was done because the readouts of the mouse experiments in this study were clear-cut (body weight loss, death). No mice were excluded from scoring.

**Cell cultures.** HEK293T (ATCC CRL-3216) cells were obtained from ATCC. Cell lines were examined for mycoplasma contamination using a Mycoplasma Detection Kit (R&D). Primary mouse T cells and human CD14$^+$ cells were cultured in RPMI-1640 medium supplemented with 10% fetal calf serum (Invitrogen Life Technology), 10 mM glutamine, 100 U ml$^{-1}$ penicillin, 100 μg ml$^{-1}$ streptomycin, and 50 μM 2-mercaptoethanol (referred to as complete RPMI medium). Human Treg cells were cultured in X-VIVO 15 medium with supplements identical to complete RPMI medium (complete X-VIVO15 medium). HEK293T cells were cultured in DMEM medium with the same supplements as for complete RPMI medium.

**Mouse tTreg cell isolation and iTreg cell differentiation.** Mouse total T cells were isolated from spleen and lymph nodes using anti-mouse Ig panning. T cells were cultured in complete RPMI 1640 medium. Mouse tTreg cells were purified from total T cells by sorting on a MoFlo Astrios system (Beckman Coulter) or by using a MACS® CD4$^+$CD25$^+$ Regulatory T Cell Isolation Kit (Miltenyi Biotech, Germany). The purity of tTreg cells was confirmed by having CD25 expression of 100% and Foxp3 expression >98.6% (Supplementary Figure 1). For WT and $Xiap^{-/-}$ mice carrying Foxp3-GFP, tTreg cells were isolated by sorting based on GFP$^+$. IL-10 and TGF-β secretion from tTreg cells were determined by a mouse IL-10 DuoSet ELISA System (R&D) and a TGF-β OptEIA Set (BD Bioscience) after tTreg cells were stimulated with immobilized anti-CD3 (4 μg ml$^{-1}$) and anti-CD28 (2 μg ml$^{-1}$) for 96 h. Naive CD4$^+$ T cells (CD4$^+$CD25$^-$CD62L$^+$CD44$^-$) were also isolated by sorting. Naive CD4$^+$ T cells were differentiated into iTreg cells by stimulation with plate-bound anti-CD3 (4 μg ml$^{-1}$) plus anti-CD28 (2 μg ml$^{-1}$) in the presence of IL-2 (20 ng ml$^{-1}$) and TGF-β (5 ng ml$^{-1}$). Three days after differentiation, iTreg cells were collected and sorted by CD25 expression. Purified iTreg cells were used for in vitro suppressive assays or rested in complete RPMI medium for 3 days before using in plasticity analysis.

**In vitro Treg cell suppression assay.** Teff (4 × 10$^5$) and antigen-presenting cells (APC, mitomycin C-treated T-depleted splenic cells, 1.2×10$^6$) were co-cultured with tTreg cells or iTreg cells in 200 μl complete RPMI medium containing 2 μg ml$^{-1}$ soluble anti-CD3 in U-bottom 96-well plates. The ratios of Treg cells and Teff were 0:1, 0.1:1, 1:2, 1:4, and 1:8. After 72 h, 0.5 μCi H$^3$-thymidine was added into culture medium for 6 h and Teff proliferation was determined. CFSE halving was used for the human Treg cells suppressive assay[66]. Human Teff were labeled with 5 μM CFSE and incubated with different amounts of Treg cells (as described above). The intensity of CFSE in CD4$^+$ T cells was analyzed by flow cytometry at 72 h.

**In vivo Treg cell suppression assay.** CD4$^+$CD25$^-$ T cells (4 × 10$^5$) from CD45.1$^+$ female mice were administered by intraperitoneal injection into CD45.1$^+$ $Rag1^{-/-}$ mice (8–10-weeks-old) with or without 1 × 10$^5$ tTreg cells from CD45.2 WT or

$Xiap^{-/-}$ tTreg cells female mice. Body weight and colitis were assessed every 3 days. After 4 weeks of T cell transfer, mice were killed and colons were removed, cleaned by PBS, fixed in 4% paraformaldehyde, and then embedded in paraffin. Sections were stained with hematoxylin and eosin (H&E). $CD45.2^+$ T cells from spleen and lymph nodes were isolated and reactivated by TPA/A23187 for 6 h and stained intracellularly for the expression of Foxp3, IFN-$\gamma$, IL-17A, and IL-6.

**Human peripheral blood mononuclear cell and T cell isolation**. Expired human white blood cell (WBC) concentrates were obtained from the Taipei Blood Bank with approval from the institutional review boards of Taipei Blood Bank and Academia Sinica. WBC concentrates were diluted with $1 \times$ HBSS, overlaid on Ficoll-Paque, and centrifuged at $400 \times g$ for 20 min. The interphase cells were collected and washed with 5% complete RPMI medium. The obtained peripheral blood mononuclear cells (PBMCs) were re-suspended in MACS® buffer (PBS containing 0.5% FBS and 2 mM EDTA). Naive human $CD4^+$ T cells ($CD4^+CD25^-$ $CD45RA^+CD45RO^-$) were isolated from human PBMCs by sorting on a MoFlo Astrios system. Human tTreg cells ($CD4^+CD127^{low}CD25^+$) and Teff ($CD4^+$ $CD25^-$) cells were similarly isolated from PBMCs. Naive human T cells were cultured in complete RPMI medium. Human tTreg cells were cultured in X-VIVO 15 medium with supplements identical to complete RPMI medium (referred to as complete X-VIVO15 medium).

**XIAP knockdown in human T cells**. The XIAP knockdown lentiviral construct was generated by cloning human XIAP-specific shRNA (5'-CCA-GAATGGTCAGTACAAA-3') into pLKO.3-Thy1.1 vector. Lentiviruses were harvested from the culture supernatant of HEK293T cells transfected with 10 μg pLKO.3-Thy1.1 or pLKO.3-Thy1.1-hXIAP shRNA, 7.5 μg psPAX2 and 3 μg pMD2G. Human naive T cells and tTreg cells were activated for 24 h by plate-bound anti-CD3 (2 μg ml$^{-1}$) and anti-CD28 (2 μg ml$^{-1}$), infected with recombinant lentivirus, and the Thy1.1-expressing cells were isolated by MACS® Beads 96 h later[67]. T cells were rested for 8 days before re-isolation of Thy1.1$^+$ T cells by MACS®. IL-2 (50 ng ml$^{-1}$) was present during the activation, infection and resting of T cells. Flow cytometry confirmed that human naive T cells remained CD4$^+$CD25$^-$ after infection and resting, and were used for iTreg cell differentiation.

**Human dendritic cell differentiation**. Human $CD14^+$ cells were isolated from PBMCs by CD14 microbeads on a MACS® Separator (Miltenyi Biotec), and were differentiated into dendritic cells (DCs) in complete RPMI medium containing GM-CSF (100 ng ml$^{-1}$) and IL-4 (50 ng ml$^{-1}$) for 7 days, with medium being replenished every 3 days. DCs were then activated by 10 ng ml$^{-1}$ TNF for 1 day and the mature DCs were used as antigen-presenting cells for human Treg cells differentiation and activation.

**Transient transfection**. HEK293T cells were transfected with plasmid DNA by Maestrofectin (MaestroGen). Cells ($4 \times 10^4$) were seeded onto 10 cm Petri-dishes overnight and transfected with 1 μg DNA combined with 2 μl transfection reagent. Cells were collected and lysed by 200 μl WCE lysis buffer (25 mM pH 7.7 HEPES, 300 mM NaCl, 0.1% Triton X-100, 1.5 mM MgCl$_2$, 0.2 mM EDTA, 0.1 mM Na$_3$VO$_4$, 50 mM NaF, 0.5 mM DTT and 10% glycerol) 24 h after transfection. HEK293T cells were grown in Dulbecco's Modified Eagle Medium (DMEM) with the same supplements as for complete RPMI medium (complete DMEM).

**Cell lysates and immunoprecipitation**. Stimulated cells used for phosphorylation analysis were lysed by p-signal cell lysis buffer (20 mM pH 7.5 Tris–HCl, 150 mM NaCl, 1 mM EDTA, 1 mM EGTA, 1% Triton X-100, 2.5 mM sodium pyrophosphate, 1 mM β-glycerophosphate, 1 mM Na$_3$VO$_4$, and 1 μg ml$^{-1}$ Leupeptin). Cells used for immunoprecipitation were lysed by WCE buffer. Antibodies and total cell lysates were mixed and rotated overnight at 4 °C. Protein–antibody complex was captured by protein G magnetic beads and subjected to further analysis.

**Immunoblot**. Cell lysates and protein samples were resolved by SDS polyacrylamide gel electrophoresis, and were transferred to PVDF membrane in transfer buffer (30 mM Tris, 250 mM glycine, 1 mM EDTA, 20% methanol) at 4 °C for 2 h at 400 mA. The membrane was blocked with blocking buffer (5% non-fat milk in 10 mM Tris–HCl pH 8.0, 150 mM NaCl, 0.1% Tween 20) at room temperature for 1 h. The treated membrane was incubated with primary antibodies overnight at 4 °C, washed, and incubated with horseradish peroxidase-conjugated secondary antibodies at room temperature for 1.5 h. The membrane was developed using ECL Western blotting detection kit. The chemiluminescence detected by X-ray film. Western blot images have been cropped for presentation. Full size images are presented in Supplementary Figs. 17–20.

**Production of inflammatory cytokines by Treg cells**. tTreg cells ($3 \times 10^5$) were stimulated with plate-bound anti-CD3 (4 μg ml$^{-1}$) plus anti-CD28 (2 μg ml$^{-1}$) and IL-2 (20 ng ml$^{-1}$), with IL-12 (50 ng ml$^{-1}$), or IL-6 (50 ng ml$^{-1}$) plus IL-1 (IL-1α and IL-1β, 20 ng ml$^{-1}$). tTreg cells were collected 5 days after culture and reactivated by TPA (50 ng ml$^{-1}$) and A23187 (500 ng ml$^{-1}$) for 6 h, fixed by Foxp3 staining kit buffer (eBioscience), and expressions of CD4, Foxp3, IFN-$\gamma$, IL-

17A and IL-6 were determined by an LSRII-18P flow cytometry system (BD Biosciences). For inhibition of Treg cells conversion, anti-IL-6R (50 μg ml$^{-1}$), anti-TNF (50 μg ml$^{-1}$), or anti-IL-1R (50 μg ml$^{-1}$), or anti-IFN-$\gamma$ (50 μg ml$^{-1}$) was included in the cell cultures. For iTreg cells, $3 \times 10^5$ iTreg cells were cultured in 12-well plates and the stimulation conditions for iTreg cells were the same as for tTreg cells. Five days after stimulation, iTreg cells were divided into two parts: one part was analyzed for intracellular expression of Foxp3 and inflammatory cytokines as in tTreg cells, and the second part was analyzed for cytokine secretion. iTreg cells ($1 \times 10^5$) in 96 well-plates were reactivated by TPA/A23187 for 18 h and cytokines in culture supernatants were measured by ELISA.

**FACS analysis**. Immune cells were labeled with specific fluorescence-conjugated antibodies. Stained cells were analyzed on a FACS LSRII (BD Biosciences), and the data collected were analyzed using FlowJo software (Flow Jo LLC, Ashland, OR). The sorting of selective cell population was performed on either FACSAria II SORP (BD Biosciences) or MoFlo Astrios (Beckman Coulter, Brea, CA). FACS gating/sorting strategies are presented in Supplementary Figure 21.

**Recombinant XIAP, XIAPΔRing and SOCS1-HA**. HEK293T cells were transfected with XIAP-Flag or XIAPΔRing-Flag or SOCS1-HA plasmids. Cells were collected and lysed by WCE buffer 48 h after transfection. Total cell lysates were incubated with protein G Mag beads and anti-Flag (M2) or anti-HA overnight at 4 °C. Recombinant XIAP-Flag and XIAPΔRing were eluted from beads by Flag peptides and were concentrated by a Vivaspin500 column (GE). SOCS1-HA-Mag beads were used directly in the ubiquitination assay.

**Ubiquitination of SOCS1 in vivo**. XIAP-pIRES1a (2 μg), SOCS1-pCDNA4 (4 μg), and EloninB/C-pcDNA4 (0.7 μg) were co-transfected with ET Ub-pRK5 (1 μg), K63 Ub-pRK5 (0.6 μg), or K48 Ub-pRK5 (0.7 μg) into HEK293T cells. Total cell lysates were prepared 24 h after transfection and were immunoprecipitated with anti-Myc.

**In vitro ubiquitination assay**. The reaction mixture containing 1 μg E1, 0.2 μg E2 (UBC13), 1 μg ubiquitin-K63, 0.1 μg E3 (XIAP or XIAPΔRing), 0.1 μg substrate (SOCS1-HA-Mag beads), 0.1 μg recombinant hEloB and hEloC in reaction buffer (25 mM Tris–HCl pH 7.5, 50 mM NaCl, 10 mM MgCl$_2$, 2 mM ATP, and 0.5 mM DTT) in a final volume of 40 μl was incubated at 30 °C for 1 h. A 4 μl sample was saved to verify the component proteins after the reaction. The complex on SOCS1-HA-Mag beads was pulled down in a magnetic rack. The extent of ubiquitination on SOCS1 was analyzed by western blotting.

**Generation of GFPRV retroviral supernatants**. SOCS1-HA was digested from SOCS1-6HA-pcDNA4 with BamHI and PmeI, and then subcloned into GFPRV to generate GFPRV-SOCS1-HA. GFPRV or GFPRV-SOCS1-HA (5 μg) was cotransfected with 3 μg psPAX2 and 2 μg pMD2G into HEK293T cells. Culture medium was collected and replenished every day. The culture media collected at day 2, 3, and 4 were combined and concentrated by ultracentrifuging at 26000 rpm on a Beckman SW28 rotor for 2 h. Pellets were resuspended in 5% BSA/PBS overnight at 4 °C and stored at −80 °C.

**SOCS1 overexpression in primary tTreg and iTreg cells**. tTreg cells ($2 \times 10^6$) were activated by plate-bound anti-CD3/CD28 (4/2 μg ml$^{-1}$) and 50 ng ml$^{-1}$ IL-2 in 12-well plates for 2 days. After activation, $1 \times 10^6$ tTreg cells were infected with retrovirus plus polybrene (8 μg ml$^{-1}$) in 24-well plates by spinning at 2000 rpm for 1 h at room temperature. Infected tTreg cells were cultured in complete RPMI medium with IL-2 (50 ng ml$^{-1}$) for 3 days. GFP$^+$CD25$^+$ cells were sorted by FACSAriaII SORP and were cultured in complete RPMI medium containing IL-2 (50 ng ml$^{-1}$) for 1 day before being subjected to a conversion assay. For the expression of SOCS1 in iTreg cells, $3 \times 10^6$ naive T cells were differentiated by plate-bound anti-CD3/CD28 (4/2 μg ml$^{-1}$), IL-2 (50 ng ml$^{-1}$) and human TGF-$\beta$ (5 ng ml$^{-1}$) in 6-well plates for 2 days. After differentiation, $2 \times 10^6$ iTreg cells in 12-well plates were infected, as described for tTreg cells infection conditions.

**Yeast preparation and quantitation**. Candida albicans (C. albicans, ATCC90028) was cultured on yeast-mold (YM) plates at 25 °C for 2 days and a single colony was inoculated into 10 ml YM broth at 30 °C for 18–24 h. C. albicans was harvested by centrifugation at 3000 rpm for 5 min and washed with 10 ml PBS twice. The yeast pellet was resuspended in PBS and the optical density was determined. For determination of fungal load in vivo, organs were removed from the infected mice, grinded and resuspended in water. The samples were serially diluted and the diluents were spread onto YM plates. The number of colonies on plates was counted after incubation at 30 °C for 2 days.

**Quantitation of cytokines in serum**. Serum was collected at the indicated time-points after C. albicans infection. Inflammatory cytokines in serum were detected by a Cytometric Beads Array Mouse Inflammatory Kit (552364, BD Biosciences). Capture Beads were mixed and added to 25 μl serum or kit-supplied standards and

incubated with PE Detection Reagent (BD Biosciences) in the dark at room temperature. The mixtures were washed and the fluorescence of the re-suspended beads was determined in a FACSCalibur system (BD Biosciences). The acquired data were analyzed with FCAP Array software (BD Biosciences).

**Statistics**. Data in this study were randomly collected but were not blinded. No data were excluded in this study. Our data mostly meet the assumption of the tests (normal distribution). GraphPad Prism 5 and Microsoft Office Excel and were used for data analysis. Unpaired two-tailed Student $t$-tests were used to compare results from between two groups. Data were presented as mean with standard deviation (s. d.) or standard error of the mean (s.e.m.), as indicated in the figure legend. Weight loss was analyzed by two-way ANOVA for multiple comparisons. Survival curves were plotted with Kaplan–Meier survival curve and analyzed by the log rank test (Mantel-cox). $P$ values <0.05 were considered significant.

**Data availability**. The authors declare that the data supporting the findings of this study are available within the article and its Supplementary files, or are available from the authors upon request.

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

## Acknowledgements

We thank Dr. David L. Vaux for the *Xiap*⁻/⁻ mice, Yamin Lin and the Institute of Molecular Biology Academia Sinica (IMB) FACS Core for cell sorting, Sue-Ping Lee and the IMB Confocal Core for confocal microscopy, and Dr. John O'Brien for editing the manuscript. This work was supported by grant MOST 105-2321-B-001-065 from the Ministry of Science and Education (Taiwan, R.O.C.), and an Academia Sinica Investigator Award from Academia Sinica, Taiwan, R.O.C.

## Author contributions

W.C.H., acquisition of data, analysis and interpretation of data, study design, statistical analysis; S.T.H., data presentation, key materials support; Y.J.C, key materials support; M.Z.L., study concept and design, study supervision, drafting of the manuscript.

## Additional information

**Competing interests:** The authors declare no competing financial interests.

