## [Peer Review File · Nature Communications]

Reviewers' comments:

Reviewer #1 (Remarks to the Author):

In this manuscript, the authors report that XIAP-deficient regulatory T cells are defective in their function due to their conversion into IFN- γ , IL-6 and IL-17-producing T regulatory cells in response to IL-6. The authors provide evidence that XIAP interacts with SOCS1 (suppressor of cytokines signaling 1) increasing its stability by ubiquitination. SOCS1 has been previously showed to be critical for the inhibitory activity of Treg cells by its ability to maintain Foxp3 levels in Treg. The authors further show that transfer of wild-type Treg partially rescue C. Albicans infection-induced inflammation in *xiap*^{-/-} animals and that anti-IL6R can block and partially restore the suppressive activity of XIAP-deficient Treg.

These observations are convincing, interesting, important and novel. They provide potential new insights into the pathophysiology the XLP-2 syndrome, which is still unclear. These data could also help to design new therapies for XIAP deficiency.

Specific comments :

-The interaction between SOCS1 and XIAP is only showed in HEK cells transiently transfected with tagged XIAP and SOCS1 constructs. The authors should provide data that this interaction also occurs in vivo in T cells (or Tregs).

-The only evidence that SOCS1 and XIAP interact together in vivo is the observation that SOCS1 expression is decreased in murine *xiap*^{-/-} T cells. Is the authors observed the same decrease in T cells from humans using experiments with shXIAP ? They also should document the role of XIAP in the SOCS1 stability in human T cells.

-Have the authors tested other ubiquitination than K63 for SOCS1 (by XIAP) ? The authors should also better discuss the role of ubiquitination for SOCS1 stabilization.

-The FACS in Fig 5e and Fig. 8a showing intracellular anti-IFN- γ staining are really not convincing. We can only trust the numbers in the gates. Could the authors improve these figures ?

- The title should also be modified to better reflect the content of the manuscript. The manuscript is not focused on the "correction of XIAP mutation-mediated primary immunodeficiency" but rather on the report of a novel function of XIAP in the context of XIAP deficiency (that could be used for future therapies)

-iTregs should be defined in the abstract

-The paper of Takahashi et al. (J Immunol. 2017) showing that the inflammatory context (in which myeloid cells are highly activated) is critical to convert Treg in ex-Treg with Th1 or Th17 phenotypes should be discussed and referred by the authors.

-In the legend of Figure 4 panel a, it should be read C-terminal in place of N-terminal

Reviewer #2 (Remarks to the Author):

The paper entitled "Correction of XIAP mutation-mediated primary immunodeficiency by autologous regulatory T cells and anti-IL-6R" by Hsieh et al. described an interesting finding that *Xiap*-deficient Tregs were defective in suppressive function partly due to decreased SOCS1 expression. They found that XIAP binds SOCS1 and promotes SOCS1 degradation. As a result,

Foxp3 becomes unstable and Xiap^{-/-} Tregs prone to secrete IFN- γ and IL-17. Interestingly, IL-6/IL-6R system seems to be important for instability of Xiap^{-/-} Tregs in mice. These are important findings for the understanding of XIAP2 mutation induced inflammation. This paper is well written and easy to understand. I have a few comments.

(1) The phenotypes of Xiap^{-/-} Tregs are very similar to that of SOCS1^{-/-} Tregs. Thus, it is reasonable to think about relationship between Xiap and SOCS1. Authors showed such link by using in vitro cell system (Figs.3,4). However, association was shown in in vitro system. It is more convincing if authors can show the interaction in vivo (primary Treg cells). If co-IP is very difficult, I recommend Proximity Ligation Assay (PLA) which becomes very popular now to show protein-protein interaction in a small number of cells.

(2) Furthermore, SOCS1 restored expression can restore the defects of Xiap^{-/-} Tregs? Foxp3 becomes more stable by overexpression of SOCS1? It is ideal if such Tregs are normal in Fig.6 experiments.

(3) Fig.6,7,8 are important and surprising results. Anti-IL-6R antibody reduced IFN- γ production from Xiap^{-/-} Tregs. Did authors tried anti-IFN- γ ? What is the role of IFN- γ in this setting? Did authors examine Th17?

(4) Anti-IL-6R antibody may reduce excess inflammation by suppressing Th17 differentiation. However, this may cause higher susceptibility to infection. Authors should examine the number of *Candida albicans* to clarify the relationship between infection and inflammation.

Point-by-point reply for MS # NCOMMS-17-20162

We have modified the title of the manuscript into “*Correction of the impaired functions in XIAP-immunodeficient regulatory T cells by anti-IL-6 receptor antibody*”, in accordance with the suggestion of Reviewer 1. We have extensively revised this paper based on the referees’ suggestions and believe we have now addressed the concerns they raised. The revisions are summarized as follows:

I. Revisions made to address issues raised by Reviewer 1

Reviewer #1 (Remarks to the Author):

In this manuscript, the authors report that XIAP-deficient regulatory T cells are defective in their function due to their conversion into IFN- γ , IL-6 and IL-17-producing T regulatory cells in response to IL-6. The authors provide evidence that XIAP interacts with SOCS1 (suppressor of cytokines signaling 1) increasing its stability by ubiquitination. SOCS1 has been previously showed to be critical for the inhibitory activity of Treg cells by its ability to maintain Foxp3 levels in Treg. The authors further show that transfer of wild-type Treg partially rescue C. Albicans infection-induced inflammation in xiap-/- animals and that anti-IL6R can block and partially restore the suppressive activity of XIAP-deficient Treg. These observations are convincing, interesting, important and novel. They provide potential new insights into the pathophysiology the XLP-2 syndrome, which is still unclear. These data could also help to design new therapies for XIAP deficiency.

Specific comments:

-The interaction between SOCS1 and XIAP is only showed in HEK cells transiently transfected with tagged XIAP and SOCS1 constructs. The authors should provide data that this interaction also occurs in vivo in T cells (or Tregs).

1. We have followed the suggestion and performed the experiment to illustrate that endogenous XIAP was immunoprecipitated by anti-SOCS1 in T cells. The new data is presented in the newly added Figure 3e, and is described on page 8.

-The only evidence that SOCS1 and XIAP interact together in vivo is the observation that SOCS1 expression is decreased in murine xiap-/- T cells. Is the authors observed the same decrease in T cells from humans using experiments with shXIAP? They also should document the role of XIAP in the SOCS1 stability in human T cells.

2. We have taken the suggestion to examine SOCS1 expression and stability in XIAP-knockdown human iTregs. In shXIAP human iTregs treated with IL-2, SOCS1 expression was reduced relative to control (pLKO.3) iTregs. This result is presented in the newly added Figure 3b, and is described on page 8.

We also determined SOCS1 protein stability in XIAP-deficient human iTregs. Treatment with cycloheximide decreased SOCS1 content in shXIAP human iTregs more rapidly than control iTregs, suggesting reduced SOCS1 protein stability in shXIAP human iTregs. This result is presented in Supplementary Figure 6c, and is described on page 8.

-Have the authors tested other ubiquitination than K63 for SOCS1 (by XIAP)? The authors should also better discuss the role of ubiquitination for SOCS1 stabilization.

3. We have performed experiments to determine if XIAP promotes K48 and K63 ubiquitination of SOCS1. We found that XIAP increased K63, but not K48, poly-ubiquitination of SOCS1. We suggest that increased K63 poly-ubiquitination is a mechanism by which XIAP stabilizes SOCS1 protein. This result is included in the newly added Figure 4b, and is presented on page 9.

It is noteworthy that SOCS1 is known to promote the ubiquitination and degradation of several signaling proteins such as VAV and Jak2. Our observation that XIAP enhances K63 poly-ubiquitination of SOCS1 and that SOCS1 becomes destabilized in the absence of XIAP suggests that K63 poly-ubiquitination contributes to SOCS1 protein stability. We have now included the role of ubiquitination in SOCS1 stability in our Discussion (page 18).

-The FACS in Fig 5e and Fig. 8c showing intracellular anti-IFN-g staining are really not convincing. We can only trust the numbers in the gates. Could the authors improve these figures?

4. We have replaced both original Figure 5e and Figure 8c with new figures making IFN- γ expression in *Xiap*^{-/-} Tregs more visible.

- The title should also be modified to better reflect the content of the manuscript. The manuscript is not focused on the correction of XIAP mutation-mediated primary immunodeficiency but rather on the report of a novel function of XIAP in the context of XIAP deficiency (that could be used for future therapies)

5. We have adopted this suggestion and modified the title to “*Correction of the impaired functions in XIAP-immunodeficient regulatory T cells by anti-IL-6 receptor antibody*”.

-iTregs should be defined in the abstract

6. As suggested by Reviewer #1, we now include a definition of iTregs in the abstract.

-The paper of Takahashi et al. (J Immunol. 2017) showing that the inflammatory context (in which myeloid cells are highly activated) is critical to convert Treg in ex-Treg with Th1 or Th17 phenotypes should be discussed and referred by the authors.

7. Following the suggestion from Reviewer #1, we now include the specific reference of Takahashi et al. (J Immunol. 2017), and discuss its findings accordingly. The study of Takahashi et al. (2017) indicates that myeloid cell priming is required for *Socs1*^{-/-} Tregs to become inflammatory. We now discuss the possibility that *Xiap*^{-/-} Tregs are similarly converted into IFN- γ -producing ex-Tregs in our Discussion (page 18).

-In the legend of Figure 4 panel a, it should be read C-terminal in place of N-terminal

8. We thank the Reviewer #1 for highlighting this error. We have now corrected it.

II. Revisions made to address issues raised by Reviewer 2

Reviewer #2 (Remarks to the Author):

The paper entitled “Correction of XIAP mutation-mediated primary immunodeficiency by autologous regulatory T cells and anti-IL-6R” by Hsieh et al. described an interesting finding that *Xiap*-deficient Tregs were defective in suppressive function partly due to decreased SOCS1 expression. They found that XIAP binds SOCS1 and promotes SOCS1 degradation. As a result, Foxp3 becomes unstable and *Xiap*^{-/-} Tregs prone to secrete IFN- γ and IL-17. Interestingly, IL-6/IL-6R system seems to be important for instability of *Xiap*^{-/-} Tregs in mice. These are important findings for true understanding of XIAP2 mutation induced inflammation. This paper is well written and easy to understand. I have a few comments.

(1) The phenotypes of *Xiap*^{-/-} Tregs are very similar to that of SOCS1^{-/-} Tregs. Thus, it is reasonable to think about relationship between *Xiap* and SOCS1. Authors showed such link by using in vitro cell system (Figs.3,4). However, association was shown in in vitro system. It is more convincing if authors can show the interaction in vivo (primary Treg cells). If co-IP is very difficult, I recommend Proximity Ligation Assay.

1. We have taken this suggestion made by both Reviewers 1 and 2, and performed the experiment to illustrate that endogenous XIAP was immunoprecipitated by anti-SOCS1 in T cells. The new data is presented in the newly added Figure 3e and is described on page 8.

(2) Furthermore, SOCS1 restored expression can restore the defects of *Xiap*^{-/-} Tregs? Foxp3 becomes more stable by overexpression of SOCS1? It is ideal if such Tregs are normal in Fig.6 experiments.

2. Following the reviewer’s suggestion, we expressed SOCS1 in *Xiap*^{-/-} Tregs by retroviral transduction. The results illustrate that SOCS1 re-introduction restored the inhibitory activity in *Xiap*^{-/-} iTregs (Supplementary Figure 9b). The instability of Foxp3 in *Xiap*^{-/-} tTregs was prevented by introduction of SOCS1 (Supplementary Figure 9e). In addition, SOCS1 expression inhibited IL-12-induced IFN- γ production in *Xiap*^{-/-} tTregs and *Xiap*^{-/-} iTregs (Supplementary Figure 9c and 9f). These results suggest that SOCS1 is one of the major targets of XIAP and that SOCS1 overexpression rescues the defective function of *Xiap*^{-/-} Tregs. These new results are presented in Supplementary Figure 9 and are described in our Results (page 11).

(3) Fig.6,7,8 are important and surprising results. Anti-IL-6R antibody reduced IFN- γ production from *Xiap*^{-/-} Tregs. Did authors try anti-IFN- γ ? What is the role of IFN- γ in this setting? Did authors examine Th17?

3.1. We have taken the suggestion to include anti-IFN- γ in the *Xiap*^{-/-} Treg conversion analysis. The results illustrate that anti-IFN- γ only weakly decreased IL-12-promoted IFN- γ expression, and it had no effect on IL-1/IL-6-induced IL-17 production in *Xiap*^{-/-} Tregs. The poor antagonistic effect of anti-IFN- γ further strengthens the specific effect of anti-IL-6R. These results are presented in the newly added Supplementary Figure 15 and are described in our Results (page 13).

3.2. In terms of the role of IFN- γ in this setting, we propose that IFN- γ represents an inflammatory conversion of Tregs. IFN- γ contributes to the inflammatory phenotype observed in *Xiap*^{-/-} mice that have encountered infection or when *Xiap*^{-/-} Tregs were transferred. However, the priming of Tregs into IFN- γ production is likely not dependent on IFN- γ . We have included these points in our Discussion (pages 17 and 18).

3.3. Reviewer 2 has raised an interesting issue on the effect of anti-IL-6R on Th17 generation, especially since IL-6 is important in priming Th17 responses and Th17 responses are known to be critical for the control of *C. albicans* infection. We measured Th17 generation in the various treatments following *C. albicans* infection. We found that *C. albicans*-induced Th17 generation was not reduced by the presence of anti-IL-6R or anti-IL-6R plus *Xiap*^{-/-} Tregs. We speculate that a single dose of anti-IL-6R does not apparently affect the priming of Th17 cells during *C. albicans* infection. This result is presented in the newly added Figure 8e, which is described in our Results (page 15) and discussed on pages 19-20 in our Discussion.

(4) Anti-IL-6R antibody may reduce excess inflammation by suppressing Th17 differentiation. However, this may cause higher susceptibility to infection. Authors should examine the number of *Candida albicans* to clarify the relationship between infection and inflammation.

4. Reviewer 2 has highlighted another critical issue regarding the effect of anti-IL-6R on the susceptibility to infection. We have examined the effect of anti-IL-6R on *C. albicans* load. The results show that anti-IL-6R did not increase fungal load in *Xiap*^{-/-} mice, either alone or in combination with *Xiap*^{-/-} Tregs. Instead, a decrease in fungal load was found in treatments containing anti-IL-6R. We also detected a comparable reduction in inflammation, as measured by neutrophil infiltration, in these treatments. Since a single dose of anti-IL-6R was administered 2 days after *C. albicans* infection in the present experiment, there is a possibility that a single dose of anti-IL-6R acts to prevent the conversion of *Xiap*^{-/-} Tregs, but does not affect priming of Th17 cells. These results are presented in the newly added Fig. 8e and 8f, which are described in our Results (Page 15) and discussed on pages 19 and 20 in our Discussion.

REVIEWERS' COMMENTS:

Reviewer #1 (Remarks to the Author):

The authors have satisfactorily replied to my comments.
I have no additional comments.

Reviewer #2 (Remarks to the Author):

Authors responded to my comments and also to another reviewer's properly. I can recommend publication of this paper in Nature Communications.

Point-by-point reply for MS # NCOMMS-17-20162A

REVIEWERS' COMMENTS:

Reviewer #1 (Remarks to the Author):

The authors have satisfactorily replied to my comments.
I have no additional comments.

Reviewer #2 (Remarks to the Author):

Authors responded to my comments and also to another reviewer's properly. I can recommend publication of this paper in Nature Communications.

We thank both reviewers for their support of this study.